# DIVERSIFIED MULTINOMIAL LOGIT CONTEXTUAL BANDITS

**Heesang Ann**
Seoul National University
sang3798@snu.ac.kr

**Taehyun Hwang**
Seoul National University
th.hwang@snu.ac.kr

**Min-hwan Oh**
Seoul National University
minoh@snu.ac.kr

## ABSTRACT

Existing contextual multinomial logit (MNL) bandits model relevance-driven choice but ignore the potential benefits of within-assortment diversity, while submodular/combinatorial bandits encode diversity in rewards but lack structured choice probabilities. We bridge this gap with the *diversified multinomial logit* (DMNL) contextual bandit, which augments MNL choice probabilities with a generally submodular diversity function, thereby formalizing the relevance–diversity trade-off within a single model. Incorporating diversity renders exact MNL assortment optimization intractable. We propose a *white-box* UCB-based algorithm, `OFU-DMNL`, that constructs assortments item-wise by maximizing optimistic marginal gains, avoids black-box optimization oracles. We show that `OFU-DMNL` achieves at least a $(1 - \frac{1}{e+1})$-*approximate* regret bound $\widetilde{\mathcal{O}}\big(d\sqrt{T/K}\big)$, where $d$ is the context dimension, $K$ the maximum assortment size, and $T$ the horizon, and attains an improved approximation factor over standard submodular baselines. Experiments demonstrate consistent gains and, relative to exhaustive enumeration, comparable regret with substantially lower runtime. Overall, DMNL bandits provide a principled and practical foundation for diversity-aware assortment optimization under uncertainty, and `OFU-DMNL` offers a statistically and computationally efficient solution.

## 1 INTRODUCTION

Sequential *assortment* selection arises whenever a platform repeatedly presents a set of items and observes a user response. E-commerce websites curate product slates, streaming services recommend a set of movies, and app stores surface a collection of apps. In each round, the decision-making agent chooses an assortment subject to a size constraint, the user selects at most one item (or makes no selection), and the agent updates future assortments based on the observed feedback. Because user preferences are not known a priori and must be learned from interactions with users, the problem is naturally cast as an online learning task: maximize cumulative reward while balancing exploration and exploitation.

A key ingredient in this setting is a probabilistic choice model that links an offered assortment to the user's selection. The *multinomial logit* (MNL) model (McFadden et al., 1978) has served as a canonical choice model for dynamic assortment learning: it represents choice probabilities through latent item utilities based on relevance, a structure that supports tractable assortment optimization and clean statistical learning guarantees. These advantages have motivated a substantial literature on MNL assortment bandits (Rusmevichientong et al., 2010; Sauré & Zeevi, 2013; Agrawal et al., 2017; 2019) and contextual variants that exploit user and item features to generalize across contexts (Cheung & Simchi-Levi, 2017; Ou et al., 2018; Chen et al., 2020; Oh & Iyengar, 2019; 2021; Perivier & Goyal, 2022; Zhang & Sugiyama, 2024; Lee & Oh, 2024; 2025). In these models, uncertainty resides in the relevance-dependent utilities, and the agent's task is to estimate them online efficiently enough to enable near-optimal sequential assortments.

However, practical assortment design is rarely driven by relevance alone: *diversity within the offered set* is often important in practice. Users tend to value assortments that span complementary attributes (e.g., different genres, brands, or styles), while assortments filled with near-duplicates can cannibalize one another and provide little additional benefit beyond offering a single represen-

tative item. At the same time, diversity is not a substitute for relevance: a diverse but irrelevant assortment still performs poorly. This creates an inherent *relevance–diversity* trade-off. Existing MNL bandits, both contextual and non-contextual, do not capture this trade-off because under the standard MNL model choice probabilities depend on items only through their individual utilities; consequently, within-assortment interactions—such as similarity-induced cannibalization or complementarity effects—are not explicitly modeled.

A natural way to incorporate such within-assortment interactions is to model the payoff of an offered set directly through a *submodular* objective, which captures diminishing returns and encourages coverage and diversity. This idea underlies a literature on submodular and combinatorial bandits (Yue & Guestrin, 2011; Chen et al., 2013; Qin et al., 2014; Chen et al., 2016; 2017; Hiranandani et al., 2020; Hwang et al., 2023), where the agent selects a subset of items each round and receives a reward specified by an (often monotone) submodular set function, possibly depending on context. While these models capture diversity-aware set selection, they abstract away the *choice-based* feedback mechanism central to assortment settings: the reward is defined as a set function rather than arising from a user selecting at most one item according to a structured discrete choice model such as MNL. Moreover, they typically do not couple relevance-driven utilities with diversity effects within a single probabilistic choice model. Consequently, there remains a modeling gap between diversity-aware set selection and relevance-based, choice-model-driven dynamic assortment learning.

We close this gap by introducing a practically motivated bandit model that embeds diversity directly into MNL choice probabilities. A key technical challenge is that, once diversity is incorporated, the tractable exact optimization available in classical MNL assortment problems is no longer applicable—the optimal assortment may require exhaustive search. Many combinatorial bandit approaches (Chen et al., 2013; Qin et al., 2014; Chen et al., 2016; Li et al., 2016; Hwang et al., 2023) circumvent this difficulty by *assuming* access to a black-box combinatorial optimization oracle with a prescribed approximation factor, an assumption that can be unrealistic in practice and that obscures the source of approximation. Our goal, instead, is to design an algorithm that simultaneously guarantees sublinear regret and a provable approximation factor *without* relying on such oracles. To this end, we introduce the diversified multinomial logit (DMNL) contextual bandit together with an efficient learning algorithm and end-to-end guarantees.

We summarize our main contributions as follows:

- **Novel assortment bandit model.** We introduce a new sequential decision-making model, which we call the *diversified multinomial logit (DMNL) contextual bandit*. In this model, user choice follows the multinomial logit choice model augmented with a diversity function—assumed submodular in general—that scores the diversity of assortments (Definition 1). To our knowledge, existing MNL bandit work does not account for assortment diversity. This is the first model to incorporate diversity directly into the choice probabilities. The model captures practical scenarios in which greater within-assortment diversity increases the likelihood of selecting an individual item for a given level of relevance, while the choice probabilities still depend on item relevance. Hence, the model addresses the natural tension between relevance and diversity—a phenomenon commonly observed in real-world recommender systems.

- **Algorithmic design.** We propose an upper confidence bound (UCB) algorithm, `OFU-DMNL` (Algorithm 1), for DMNL bandits. The salient feature of the algorithm is an item-wise optimistic construction of an assortment: it incrementally adds the item that yields the largest marginal increase in the optimistic reward estimate. This process is computationally efficient and comes with a provable guarantee on an *approximation* rate.[1]

- **Regret guarantee.** We prove that our proposed algorithm is statistically efficient. Under a sufficient condition on the diversity function (Definition 2), we prove that the algorithm achieves at least a $(1-\frac{1}{e+1})$-approximate regret bound of $\widetilde{\mathcal{O}}(d\sqrt{T/K})$ (Theorem 3), where $d$ is the context feature dimension, $K$ is the maximum assortment size, $T$ is the total number

---

[1]Due to the augmented diversity function, exact assortment optimization is no longer tractable as in prior work (Ou et al., 2018; Oh & Iyengar, 2019; 2021; Perivier & Goyal, 2022; Lee & Oh, 2024; 2025); hence, we must resort to approximation. Unlike existing work on combinatorial bandits, which assumes access to a black-box optimization oracle returning a super-arm (a set of base arms) with a prescribed approximation factor, our algorithm employs a transparent, white-box construction for which we directly prove the approximation rate.

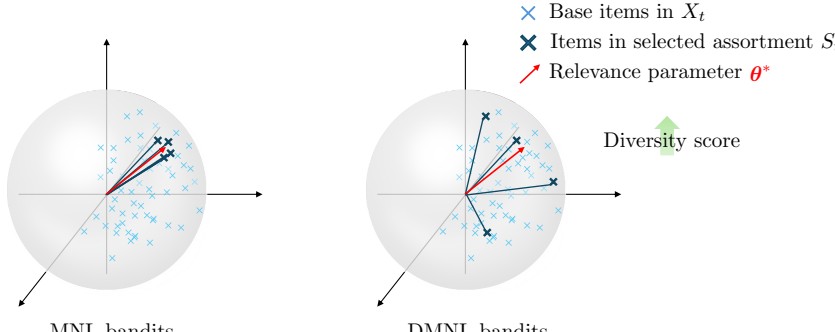

Figure 1: Behavioral differences between MNL bandit algorithms and the proposed DMNL bandit algorithms ($d = 3, K = 4$). While MNL bandit algorithms select the top-$K$ relevant items in the uniform revenue setting, DMNL bandit algorithms consider both item relevance and assortment diversity, resulting in more diverse selections whose degree depends on the DMNL environment.

of rounds, and $\widetilde{\mathcal{O}}$ suppresses logarithmic factors. This bound closely matches that of nearly minimax-optimal algorithms for MNL bandits under uniform revenues (Lee & Oh, 2024), despite additionally learning a diversity parameter.

- **Approximation guarantee.** We show that the item-wise greedy construction under the DMNL model attains a stronger approximation rate (Theorem 1) than those established in the submodular maximization literature (Nemhauser et al., 1978; Feige, 1998; Yue & Guestrin, 2011). Unlike the prior literature on MNL bandits—where identifying the optimal assortment can be done efficiently—finding the optimal assortment in DMNL bandits while accounting for diversity requires exhaustive search, making approximation guarantees essential. By leveraging the MNL structure together with the submodularity of the diversity function, we obtain an improved approximation rate of at least $\left(1 - \frac{1}{e+1}\right)$, without having to rely on black-box optimization oracles.

- **Numerical performance.** Extensive numerical experiments show that our algorithm outperforms benchmark methods across a wide range of scenarios. In particular, even relative to exhaustive enumeration over all possible assortments, our algorithm achieves comparable regret while offering a substantial reduction in running time. Hence, our proposed method is both computationally and statistically efficient.

## 2 PRELIMINARIES

### 2.1 NOTATIONS AND DEFINITIONS

We use $\|\mathbf{x}\|_2$ to denote the $l_2$-norm of a vector $\mathbf{x} \in \mathbb{R}^d$ and $\|\mathbf{x}\|_{\mathbf{A}} := \sqrt{\mathbf{x}^\top \mathbf{A} \mathbf{x}}$ to denote the weighted norm of $\mathbf{x}$ induced by a positive definite matrix $\mathbf{A} \in \mathbb{R}^{d \times d}$. For a symmetric matrices $\mathbf{V}$ and $\mathbf{W}$ of the same dimensions, $\mathbf{V} \succeq \mathbf{W}$ means that $\mathbf{V} - \mathbf{W}$ is positive semi-definite. For a positive integer $n$, we denote by $[n]$ the set $\{1, \dots, n\}$. Refer to Appendix A for additional notation.

### 2.2 PROBLEM SETTING

**Diversified Multinomial Logit (DMNL) Contextual Bandits.** We consider a sequential assortment selection problem where, in each round $t \in [T]$, the agent receives a set of feature vectors $X_t := \{\mathbf{x}_{t1}, \dots, \mathbf{x}_{tN}\} \subset \mathbb{R}^d$, which may be chosen adversarially. The agent then offers an assortment of size of at most $K$, i.e., $S_t = \{i_1, \dots, i_l\} \in \mathcal{S} := \{S \subset [N] : |S| \le K\}$, where $l \le K$. After presenting the assortment $S_t$, the agent observes the user's decision $i_t \in S_t \cup \{0\}$, where 0 represents the "outside option", indicating that the user does not choose any item from $S_t$. The user's selection $i_t$ is modeled by the MNL choice model (McFadden et al., 1978).

In the existing MNL bandit framework (Cheung & Simchi-Levi, 2017; Ou et al., 2018; Oh & Iyengar, 2019; 2021; Chen et al., 2020; Lee & Oh, 2024; 2025) the click probability that a user selects

an item depends only on the relevance utility of the item and the other items in the assortment. We instead consider an MNL choice model that incorporates the diversity of the assortment.

**Definition 1** (Diversified multinomial logit choice model). *For each round $t \in [T]$, let $g_t : \mathcal{S} \to \mathbb{R}_{\geq 0}$ be a given monotone submodular function, where $g_t(S)$ quantifies the diversity of the items in the assortment $S$ in round $t$. Then, the probability of selecting an item $i_t \in S_t \cup \{0\}$ in round $t$ is defined as follows:*

$$
\begin{aligned}
\mathbb{P}(i_t = i \mid X_t, S_t) =: p_t(i \mid S_t, \boldsymbol{\theta}^*, \lambda^*) &:= \frac{\exp(\mathbf{x}_{ti}^\top \boldsymbol{\theta}^*)}{\exp(-\lambda^* g_t(S_t)) + \sum_{j \in S_t} \exp(\mathbf{x}_{tj}^\top \boldsymbol{\theta}^*)} \,, \\
\mathbb{P}(i_t = 0 \mid X_t, S_t) =: p_t(0 \mid S_t, \boldsymbol{\theta}^*, \lambda^*) &:= \frac{\exp(-\lambda^* g(S_t))}{\exp(-\lambda^* g_t(S_t)) + \sum_{j \in S_t} \exp(\mathbf{x}_{tj}^\top \boldsymbol{\theta}^*)} \,,
\end{aligned}
\tag{1}
$$

*where $\boldsymbol{\theta}^* \in \mathbb{R}^d$ and $\lambda^* \in \mathbb{R}$ are unknown parameters that represent the degree of relevance and diversity, respectively.*

**Remark 1.** *Inspired by the submodular bandit literature (Yue & Guestrin, 2011; Chen et al., 2017; Hiranandani et al., 2020), our model captures diversity through monotone submodular functions $g_t$ (Definition 3 and 4). Common notions of diversity—such as counting the number of distinct categories, measuring coverage of item attributes (e.g., brands or genres), or quantifying dispersion in an embedding space via pairwise distances or spectral properties of a Gram matrix—naturally exhibit diminishing diversity returns: adding an item similar to those already selected contributes less than adding one from a new category or a distant region in feature space. Such measures are monotone and submodular by construction, so monotone submodular functions provide a unifying and behaviorally plausible abstraction for a broad class of real-world diversity notions.*

In DMNL bandit setting, the user choice follows the DMNL model. In other words, the choice feedback $\mathbf{y}_t := (y_{t0}, y_{t1}, \ldots, y_{tl})$ follows the following MNL distribution:

$$
\mathbf{y}_t \sim \text{Multinomial}\{1, (p_t(0 \mid S_t, \boldsymbol{\theta}^*, \lambda^*), \ldots, p_t(i_l \mid S_t, \boldsymbol{\theta}^*, \lambda^*))\} \,,
$$

where the parameter 1 indicates that $\mathbf{y}_t$ is a single-trial sample, i.e. $y_{t0} + \sum_{k=1}^{l} y_{tk} = 1$.

When two assortments consist of items with identical utility values, the one with a higher diversity score reduces the probability of the outside option being chosen. As a result, the probability of selecting each item in the assortment increases, leading to a higher expected reward for the assortment. Conversely, if an assortment with a lower diversity score is offered, the outside option becomes more attractive, resulting in lower selection probabilities for the items in the assortment.

**Remark 2.** *We note that the proposed DMNL model generalizes the existing MNL models (Cheung & Simchi-Levi, 2017; Ou et al., 2018; Oh & Iyengar, 2019; Chen et al., 2020; Oh & Iyengar, 2021; Lee & Oh, 2024; 2025). When the diversity function $g_t(S)$ is constant across all assortments, the DMNL model reduces to the existing MNL model. In contrast, the existing MNL model does not allow the outside option's attraction to vary with the offered set, as DMNL does.*

Then, the expected reward of an assortment $S$ in round $t$ is defined as follows:

$$
R_t(S, \boldsymbol{\theta}^*, \lambda^*) := \sum_{i \in S} p_t(i \mid S, \boldsymbol{\theta}^*, \lambda^*) = \sum_{i \in S} \frac{\exp(\mathbf{x}_{ti}^\top \boldsymbol{\theta}^*)}{\exp(-\lambda^* g_t(S)) + \sum_{j \in S} \exp(\mathbf{x}_{tj}^\top \boldsymbol{\theta}^*)} \,.
$$

The goal of the agent is to maximize the total expected reward, or equivalently, to minimize the cumulative regret over $T$ rounds, defined as total difference in expected reward between the offline optimal assortment $S_t^* = \text{argmax}_{S \in \mathcal{S}} R_t(S, \boldsymbol{\theta}^*, \lambda^*)$ and the assortment $S_t$ offered by the agent.

**Remark 3.** *Previous works on MNL bandits (Oh & Iyengar, 2019; Chen et al., 2020; Oh & Iyengar, 2021; Zhang & Luo, 2024; Lee & Oh, 2024; 2025) have also studied the non-uniform revenue setting, where in each round, the agent observes the item-wise revenues $\{r_{ti}\}_{i=1}^{N}$. In this setting, the optimal assortment is heavily influenced by high-revenue items, making the diversity of the assortment less critical to the reward. In other words, encouraging diversity in the selected assortment may not align with the objective of reward maximization under non-uniform revenue. By contrast, in the uniform revenue setting ($r_{ti} = 1$) we study, maximizing diversity directly contributes to increasing the overall click probability and expected reward, making it a more appropriate objective (Figure 1). We focus on the uniform revenue setting not only for analytical clarity but also because it allows us to isolate and rigorously study the effect of assortment diversity on user choice behavior.*

$\gamma$**-approximate Regret.** In the case of uniform revenues in MNL bandit setting, maximizing the expected reward of an assortment over all sets $S \in \mathcal{S}$ reduces to selecting the $K$ items with the highest relevance utility. However, unlike in the existing MNL bandit literature, such a top-$K$ selection strategy is no longer sufficient in the DMNL model. Because the diversity of an assortment influences click probabilities, the expected reward depends not only on individual item relevance utilities but also on the overall diversity of the selected set. Thus, finding the optimal assortment requires evaluating all $\binom{N}{K}$ subsets, which is computationally prohibitive even when $\boldsymbol{\theta}^*$ and $\lambda^*$ are known. In previous combinatorial bandit works (Chen et al., 2013; Qin et al., 2014; Chen et al., 2016; Li et al., 2016; Hwang et al., 2023; Liu et al., 2024; 2025), such computational challenges are typically addressed by assuming access to a $\gamma$-approximate oracle, and the performance of algorithms is evaluated via cumulative $\gamma$-approximate regret rather than exact regret. The $\gamma$-approximate regret at round $t$ is defined as $\mathcal{R}^{\gamma}(t, S_t) = \gamma R_t(S_t^*, \boldsymbol{\theta}^*, \lambda^*) - R_t(S_t, \boldsymbol{\theta}^*, \lambda^*)$. Then, the alternative objective of the agent is to minimize the *cumulative $\gamma$-regret*, defined as

$$\mathcal{R}^{\gamma}(T) := \sum_{t=1}^{T} \mathcal{R}^{\gamma}(t, S_t) = \sum_{t=1}^{T} \gamma R_t(S_t^*, \boldsymbol{\theta}^*, \lambda^*) - R_t(S_t, \boldsymbol{\theta}^*, \lambda^*).$$

We adopt this standard evaluation metric but *do not rely on an oracle*. Instead, in Section 3.1, we explicitly construct a computationally efficient assortment selection strategy that serves as an approximation oracle. Specifically, we show that the item-wise greedy construction (Eq.(6)) achieves a provable approximation ratio $\gamma \geq 1 - \frac{1}{e+1}$ with respect to the offline optimum, while requiring only $\mathcal{O}(NK)$ computation per round. This result enables practical deployment without sacrificing theoretical guarantees.

Following prior work on MNL contextual bandits, we make the following boundedness assumption.

**Assumption 1** (Boundedness). *We assume that $\|[\boldsymbol{\theta}^*, \lambda^*]\|_2 \leq 1$, $\|\mathbf{x}_{ti}\|_2 \leq 1$ and $0 \leq g(S_t) \leq 1$ for all $t \in [T], i \in [N]$, and there exists a constant $l > 0$ such that $l < \lambda^*$.*

The boundedness in Assumption 1 is standard in the MNL bandit literature (Oh & Iyengar, 2019; 2021; Perivier & Goyal, 2022; Zhang & Sugiyama, 2024; Lee & Oh, 2024; 2025). Since we focus on scenarios where the diversity of an assortment influences user choice behavior, we assume that the effect of diversity is strictly positive—i.e., the minimum effect of diversity is bounded below by a positive constant $l$. We note that our proposed algorithm does not require the knowledge of $l$.

## 3 MAIN RESULTS

### 3.1 APPROXIMATION GUARANTEE OF ITEM-WISE GREEDY ASSORTMENT

In this section, as an instantiation of $\gamma$-approximate oracle, we show that the item-wise greedy construction can approximate the offline optimal assortment reward $R_t(S_t^*, \boldsymbol{\theta}^*, \lambda^*)$. The item-wise greedy construction refers to a process that incrementally builds a solution by repeatedly adding the item with the highest marginal gain. To be specific, for any $k \in [K]$, the $k$-th element added during the item-wise greedy construction is:

$$a_k = \underset{a \in [N] \setminus \{a_1, \ldots, a_{k-1}\}}{\operatorname{argmax}} R_t(\{a_1, \ldots, a_{k-1}\} \cup \{a\}, \boldsymbol{\theta}^*, \lambda^*). \tag{2}$$

It is well known that if the expected reward function $R_t$ is a monotone submodular function with respect to $S \in \mathcal{S}$, the item-wise greedy construction in Eq.(2) can achieve a $(1 - \frac{1}{e})$-approximation rate (Nemhauser et al., 1978), and that obtaining an approximation rate better than $(1 - \frac{1}{e})$ is intractable (Feige, 1998).

On the other hand, since $R_t(S, \boldsymbol{\theta}^*, \lambda^*)$ increases as more items are added to $S$, it is a monotone set function (Definition 3). Moreover, by the definition of submodular functions (Definition 4), the LogSumExp function of the form $\log\left(\sum_{i \in S} \exp(\mathbf{x}_{ti}^\top \boldsymbol{\theta}^*)\right)$ is submodular. The expected reward of an assortment $S$ can be written as $R_t(S, \boldsymbol{\theta}^*, \lambda^*) = \frac{\exp(f_t(S))}{1+\exp(f_t(S))}$, where $f_t(S) := \log\left(\sum_{i \in S} \exp(\mathbf{x}_{ti}^\top \boldsymbol{\theta}^* + \lambda^* g_t(S))\right) = \log\left(\sum_{i \in S} \exp(\mathbf{x}_{ti}^\top \boldsymbol{\theta}^*)\right) + \lambda^* g_t(S)$. Since $f_t(S)$ is a non-negative sum of submodular functions, it remains submodular. Moreover, $\frac{\exp(x)}{1+\exp(x)}$ is a non-decreasing concave function for $x > 0$, and it is known that the composition of a submodular

---

**Algorithm 1** `OFU-DMNL`

---

1: **Input:** diversity function $\{g_t\}_{t \geq 1}$, regularization parameter $\Lambda$, confidence radius $\{\alpha_t\}_{t \geq 1}$, step size $\eta$, exploration parameter $\nu$
2: **Initialization:** $\mathbf{H}_1 = \Lambda \mathbf{I}_{d+1}$ and $\mathbf{w}_1$ at any point in $\mathcal{W}$.
3: **for** $t = 1, \ldots, T$ **do**
4:      **if** $\|[\mathbf{0}_d, 1]\|_{\mathbf{H}_t^{-1}} \geq \nu \frac{\hat{\lambda}_t}{\alpha_t}$ **then**
5:          Randomly choose $S_t \sim \mathrm{Unif}(\mathcal{S})$ with $|S_t| = K$
6:      **else**
7:          $S_t \leftarrow \emptyset$
8:          **for** $k = 1, \ldots, K$ **do**
9:              $a_{t,k} = \mathrm{argmax}_{a \in [N] \setminus S_t} \widetilde{R}_t(\{a_{t,1}, \ldots, a_{t,k-1}\} \cup \{a\})$
10:              $S_t \leftarrow S_t \cup \{a_{t,k}\}$
11:      Offer $S_t$ and observe $y_t$
12:      Update $\widetilde{\mathbf{H}}_t = \mathbf{H}_t + \eta \, \mathcal{G}_t(\mathbf{w}_t)$, $\mathbf{w}_{t+1}$, and $\mathbf{H}_{t+1} = \mathbf{H}_t + \mathcal{G}_t(\mathbf{w}_{t+1})$

---

function with a non-decreasing concave function preserves submodularity (Proposition 2). Therefore, $R_t(S, \boldsymbol{\theta}^*, \lambda^*)$ is also a submodular function.

Consequently, the item-wise greedy construction in Eq.(2) achieves at least a $(1 - \frac{1}{e})$ approximation rate. This approximation guarantee holds for general monotone submodular functions under cardinality constraints ($|S| \leq K$). However, we further show that by leveraging the specific structure of the MNL model, it is possible to obtain an approximation ratio that strictly improves upon the standard $(1 - \frac{1}{e})$ rate.

**Theorem 1** (Improved approximation rate for MNL submodular function). *Let $S_t^{greedy}$ be the solution computed by Eq.(2). For any $t \geq 1$, if $g_t$ is monotone and submodular, then we have*

$$R_t(S_t^{greedy}, \boldsymbol{\theta}^*, \lambda^*) \geq \frac{\psi_0(1 + \psi_0^\alpha)}{\psi_0^\alpha(1 + \psi_0)} \cdot R_t(S_t^*, \boldsymbol{\theta}^*, \lambda^*),$$

*where $\psi_0$ is a solution to the equation $x^\alpha = \alpha x + \alpha - 1$, with $\alpha = \frac{e}{e-1}$.*

Theorem 1 holds for any parameter configuration $[\boldsymbol{\theta}, \lambda] \in \mathbb{R}^{d+1}$, provided that both the item-wise greedy construction and the optimal assortment are evaluated under the same parameters. Moreover, since a crude lower bound for $\frac{\psi_0(1+\psi_0^\alpha)}{\psi_0^\alpha(1+\psi_0)}$ is $\frac{1}{e+1}$, for simplicity, we may state that the item-wise greedy construction in Eq.(2) achieves at least a $(1 - \frac{1}{e+1})$-approximate rate. This surpasses the existing $(1 - \frac{1}{e})$ approximation rate attainable under general submodularity assumption alone. The improvement arises from the structural properties of the MNL reward function, and is of standalone theoretical interest. The detailed proof is provided in Appendix D.

## 3.2 ALGORITHM

In this section, we propose `OFU-DMNL`, an algorithm that leverages the *optimism-in-the-face-of-uncertainty* (OFU) principle in estimating the unknown relevance utility and diversity parameters. The complete process is described in Algorithm 1, consisting of three primary stages.

**Diversity-augmented parameter estimation.** Let $\mathbf{z}_{ti}(S) := [\mathbf{x}_{ti}, g_t(S)] \in \mathbb{R}^{d+1}$ be a diversity-augmented feature vector, and $\mathbf{w}^* := [\boldsymbol{\theta}^*, \lambda^*] \in \mathbb{R}^{d+1}$. Then, the DMNL probability in Eq.(1) can be represented by

$$p_t(i \mid S, \boldsymbol{\theta}^*, \lambda^*) =: p_t(i \mid S, \mathbf{w}^*) = \frac{\exp(\mathbf{z}_{ti}(S)^\top \mathbf{w}^*)}{1 + \sum_{j \in S_t} \exp(\mathbf{z}_{tj}(S)^\top \mathbf{w}^*)},$$

$$p_t(0 \mid S, \boldsymbol{\theta}^*, \lambda^*) =: p_t(0 \mid S, \mathbf{w}^*) = \frac{1}{1 + \sum_{j \in S_t} \exp(\mathbf{z}_{tj}(S)^\top \mathbf{w}^*)}.$$

Consequently, parameter estimation in the DMNL model—namely, $(\boldsymbol{\theta}^*, \lambda^*)$—can be reformulated as estimating a single parameter vector $\mathbf{w}^*$ using diversity-augmented feature vectors $\mathbf{z}_{ti}(S)$,

similarly to the procedure used in existing MNL models. Adapting the computationally efficient parameter estimation used in Lee & Oh (2024), we use the online mirror descent algorithm to estimate the parameter $\mathbf{w}^*$. Let us define the multinomial logit loss function at round $t$ as $\ell_t(\mathbf{w}) := -\sum_{i \in S_t} y_{ti} \log p_t(i \mid S_t, \mathbf{w})$, and estimate the true parameter $\mathbf{w}^*$ as follows:

$$[\hat{\boldsymbol{\theta}}_{t+1}, \hat{\lambda}_{t+1}] = \mathbf{w}_{t+1} = \operatorname*{argmin}_{\mathbf{w} \in \mathcal{W}} \left\{ \langle \nabla \ell_t(\mathbf{w}_t), \mathbf{w} \rangle + \frac{1}{2\eta} \|\mathbf{w} - \mathbf{w}_t\|_{\widetilde{\mathbf{H}}_t}^2 \right\}, \quad \forall t \geq 1, \tag{3}$$

where $\mathcal{W} := \{\mathbf{w} \in \mathbb{R}^{d+1} : \|\mathbf{w}\|_2 \leq 1\}$, $\eta > 0$ is the step-size parameter, and $\widetilde{\mathbf{H}}_t := \mathbf{H}_t + \eta \, \mathcal{G}_t(\mathbf{w}_t)$, with $\mathbf{H}_t := \Lambda \mathbf{I}_{d+1} + \sum_{s=1}^{t-1} \mathcal{G}_s(\mathbf{w}_{s+1})$ and

$$\mathcal{G}_t(\mathbf{w}) = \sum_{i \in S_t} p_t(i \mid S_t, \mathbf{w}) \mathbf{z}_{ti}(S_t) \mathbf{z}_{ti}(S_t)^\top - \sum_{i \in S_t} \sum_{j \in S_t} p_t(i \mid S_t, \mathbf{w}) p_t(j \mid S_t, \mathbf{w}) \mathbf{z}_{ti}(S_t) \mathbf{z}_{tj}(S_t)^\top .$$

Based on the estimated parameter $\mathbf{w}_t$ and a suitably chosen confidence radius $\alpha_t$, we have with high probability that $\|\mathbf{w}_t - \mathbf{w}^*\|_{\mathbf{H}_t} \leq \alpha_t$ (Lemma 1 in Lee & Oh (2024)). This concentration bound enables us to construct an optimistic estimate of the diversity-augmented utility by evaluating it over the diversity-augmented feature vector as:

$$\operatorname{ucb}(\mathbf{z}_{ti}(S)) := [\mathbf{x}_{ti}, g_t(S)]^\top \mathbf{w}_t + \alpha_t \|[\mathbf{x}_{ti}, g_t(S)]\|_{\mathbf{H}_t^{-1}} . \tag{4}$$

Based on the optimistic utility estimates $\operatorname{ucb}(\mathbf{z}_{ti}(S))$, we formulate the diversified optimistic expected reward for a given assortment $S$ as:

$$\widetilde{R}_t(S) := \sum_{i \in S} \frac{\exp(\operatorname{ucb}(\mathbf{z}_{ti}(S)))}{1 + \sum_{j \in S} \exp(\operatorname{ucb}(\mathbf{z}_{tj}(S)))} . \tag{5}$$

As discussed in Section 2.2, in the existing MNL bandits with uniform revenues, it is sufficient to construct an assortment by selecting the top-$K$ items with the highest optimistic utility estimates, since such an assortment serves as an optimistic estimate of the offline optimal reward. However, in the DMNL model, the diversity-augmented feature vector $\mathbf{z}_{ti}(S)$ depends on the entire assortment $S$, which means that the optimistic reward $\widetilde{R}_t(S)$ cannot be computed by evaluating each item in isolation. As a result, identifying the assortment that maximizes $\widetilde{R}_t(S)$ requires evaluating $\binom{N}{K}$ combinations, which is computationally prohibitive for large $N$ or $K$. In the following paragraph, we introduce a computationally efficient method—serving as the main component of our algorithm—for approximating $\max_S \widetilde{R}_t(S)$ without exhaustive enumeration.

**Item-wise optimistic construction.** As discussed in Section 3.1, the item-wise greedy construction using the true parameters $[\boldsymbol{\theta}^*, \lambda^*]$ achieves at least $(1 - \frac{1}{e+1})$-approximation to the optimal assortment. However, since the true model parameters are unknown to the agent, we replace them with their estimates. In particular, we use an optimistic estimate of the expected reward to guide assortment construction, which encourages exploration over uncertain items while preserving computational efficiency. Using the diversified optimistic reward defined in Eq.(5), we apply an item-wise optimistic construction: for each $k \in [K]$,

$$a_k = \operatorname*{argmax}_{a \in [N] \setminus \{a_1, \dots, a_{k-1}\}} \widetilde{R}_t(\{a_1, \dots, a_{k-1}\} \cup \{a\}) . \tag{6}$$

This procedure mirrors the ideal greedy construction under the true model, but substitutes the unknown parameters with optimistic estimates—hence the name *item-wise optimistic construction*. The agent then offers the assortment $S_t$ obtained via Eq.(6). We note that the computation complexity of the item-wise optimistic construction in Eq.(6) is $\mathcal{O}(NK)$ for each round.

**Adaptive exploration.** As the two parameters $\boldsymbol{\theta}$ and $\lambda$ are estimated jointly via the diversity-augmented feature vector, their individual uncertainties cannot be disentangled. However, the joint confidence width may not provide a sufficiently tight uncertainty estimate for $\lambda^*$ alone. This looseness in the confidence interval may result in a failure to ensure the optimism of the item-wise optimistic construction in Eq.(6). To address this, we employ an adaptive exploration that triggers when the confidence on the diversity parameter estimate is deemed insufficient. We show that the number of such rounds is at most $\mathcal{O}(\sqrt{d} \log T)$ (Lemma 6).

## 3.3 REGRET BOUND

In this section, we establish an upper bound on the cumulative $\gamma$-approximate regret incurred by the proposed algorithm. To facilitate the theoretical analysis, we first present a set of technical assumptions under which the regret bound is derived.

**Assumption 2** (Non-degeneracy). *The feature set $X_t = \{\mathbf{x}_{t1}, \ldots, \mathbf{x}_{tN}\}$ spans $\mathbb{R}^d$ for all $t \in [T]$, $g_t$ is not a constant over $\mathcal{S}_K := \{S \subset [N] : |S| = K\}$, i.e., $\exists S, S' \in \mathcal{S}_K$ such that $g_t(S) \neq g_t(S')$.*

**Definition 2** ($\omega$-strict submodular function). *For $\omega \in (0,1)$, a submodular function $f$ is said to be $\omega$-strict submodular if and only if for every $S \subseteq S'$ with $f(S) \neq f(S')$ and every $e \notin S'$, $f$ satisfies*

$$f(S' \cup \{e\}) - f(S') \leq (1-\omega)\left(f(S \cup \{e\}) - f(S)\right).$$

**Assumption 3** (Strict submodularity). *The diversity score function $g_t$ is monotone and $\omega$-strict submodular for some $\omega > 0$.*

**Discussions of assumptions.** Assumption 2 is used to ensure the diversity-augmented feature set $\{[\mathbf{x}_{ti}, g_t(S)]\}_{i \in [N], S \in \mathcal{S}_K}$ spans $\mathbb{R}^{d+1}$. Under Assumption 2, there exist $S_1, S_2 \in \mathcal{S}_K$ such that $S_1 \cap S_2 \neq \emptyset$ and $g_t(S_1) \neq g_t(S_2)$. Let $i_0 \in S_1 \cap S_2$. Then we have $[\mathbf{0}_d, 1] = \frac{1}{g_t(S_1) - g_t(S_2)}([\mathbf{x}_{ti_0}, g_t(S_1)] - [\mathbf{x}_{ti_0}, g_t(S_2)])$. This shows that the $(d+1)$-th unit vector $[\mathbf{0}_d, 1] \in \mathbb{R}^{d+1}$ can be expresses as a linear combination of diversity-augmented features. Moreover, the first d-dimensional components of $\mathbb{R}^{d+1}$ can be spanned by the set $\{[\mathbf{x}_{ti}, 0]\}_{i \in [N]}$ due to Assumption 2. Therefore, the diversity-augmented feature set $\{[\mathbf{x}_{ti}, g_t(S)]\}_{i \in [N], S \in \mathcal{S}_K}$ spans $\mathbb{R}^{d+1}$. With the diversity-augmented feature set spanning $\mathbb{R}^{d+1}$, we can define a constant $\sigma_0 > 0$ such that for all $t \in [T]$,

$$\frac{1}{|\mathcal{S}_K| \cdot K} \sum_{S \in \mathcal{S}_K} \sum_{i \in S} [\mathbf{x}_{ti}, g_t(S)][\mathbf{x}_{ti}, g_t(S)]^\top \succeq \sigma_0 \mathbf{I}_{d+1}. \tag{7}$$

We note that this type of non-degeneracy condition is also commonly used in prior works on GLM and MNL bandits (Li et al., 2017; Chen et al., 2020; Oh & Iyengar, 2021).

The strict submodularity in Assumption 3 implies that for any $S \subsetneq S'$ and any element $e \notin S$, the marginal gain of $g_t$ from adding $e$ to $S'$ is strictly smaller than that from adding $e$ to $S$. This condition more explicitly captures the law of diminishing returns than the standard definition of submodularity (Definition 4). Unlike prior submodular bandit works (Yue & Guestrin, 2011; Chen et al., 2017; Hiranandani et al., 2020), the DMNL bandit setting assumes that the agent does not receive intermediate feedback on the diversity score during the construction of the assortment. Moreover, the agent does not observe the marginal gain in reward for each item in the assortment, which significantly increases the difficulty of learning. On the other hand, Yue & Guestrin (2011) assume access to the marginal contribution of each item after the assortment is offered, while Chen et al. (2017) receive interactive feedback on the gain of each added item during the assortment construction process. Similarly, Hiranandani et al. (2020) assume that in a cascading setting, the agent receives feedback corresponding to the utility gain of adding new items to a previously selected subset. In contrast, in the DMNL bandit setting, the agent only observes the final reward associated with the offered assortment $S_t$, making the problem significantly more challenging. However, under the strict submodularity assumption we show that it is possible to recover submodularity of $\widetilde{R}_t(S)$ after sufficient exploration, even *without intermediate feedback*. Please refer to Appendix C for a detailed discussion on strict submodularity.

We first present a lower bound for the worst-case expected regret in our DMNL bandit setting.

**Theorem 2** (Regret lower bound). *Let Assumption 1, 2, and 3 hold. Suppose $d$ is divisible by $4$ and $T \geq C \cdot d^4(K+1)^2/K$ for some constant $C > 0$. Then, in the DMNL bandit setting, for any policy $\pi$, there exists a worst-case problem instance such that the expected regret of $\pi$ is lower bounded as*

$$\sup_{\boldsymbol{\theta}, \lambda} \mathbb{E}^\pi_{\boldsymbol{\theta}, \lambda}\left[\sum_{t=1}^T R_t(S_t^*, \boldsymbol{\theta}, \lambda) - R_t(S_t, \boldsymbol{\theta}, \lambda)\right] \geq \Omega\left(d\sqrt{\frac{T}{K}}\right).$$

**Discussion of Theorem 2.** The theorem shows that the regret lower bound of our DMNL bandit setting matches that of MNL bandits under uniform revenues (Lee & Oh, 2024). In our setting,

the choice probabilities depend on the value of the assortment's diversity function, and therefore the existing lower-bound arguments for MNL bandits cannot be applied directly. Specifically, we consider a non-constant, strict submodular $g_t$ and derive an inequality for the instantaneous regret lower bound, even though the optimal assortment includes an item that is not individually optimal in terms of their relevance scores. The detailed proof is provided in Appendix E.

We now present our main result: the cumulative $\gamma$-approximate regret bound for Algorithm 1.

**Theorem 3** (Regret upper bound of OFU-DMNL). *Suppose that Assumptions 1, 2, and 3 hold. For any $\delta \in (0,1)$, if we set the algorithmic parameters in Algorithm 1 as follows: $\alpha_t = \mathcal{O}(\sqrt{d}\log t \log K)$, $\eta = \frac{1}{2}\log(K+1) + 2$, $\Lambda = 84\sqrt{(d+1)\eta}$, $\nu = \frac{\omega}{2}$, then with probability at least $1 - \delta - (d + 1)T^{-\mathcal{O}(\frac{\sigma_0 \sqrt{d}\log K}{\kappa l \omega K})}$, the cumulative $\gamma$-regret of OFU-DMNL is upper-bounded by*

$$R^\gamma(T) = \widetilde{\mathcal{O}}\left( \frac{\sqrt{K}(d+1)}{K+1} \cdot \sqrt{T} + \frac{1}{\kappa}\left( (d+1)^2 + \frac{\sqrt{d}}{l\omega} \right) \right),$$

*where $\kappa := \min_{t \in [T], S \in \mathcal{S}, i \in S, \|\boldsymbol{\theta}\|_2 \leq 1, 0 \leq \lambda \leq 1} p_t(i|S, \boldsymbol{\theta}, \lambda)p_t(0|S, \boldsymbol{\theta}, \lambda) > 0$ is a problem-dependent instance, and $\gamma \geq (1 - \frac{1}{1+e})$.*

**Discussion of Theorem 3.** The theorem establishes that the regret upper bound of Algorithm 1 is nearly minimax-optimal, as it matches the lower bound for our problem setting in its dependence on $d$, $K$, and $T$, up to the effects introduced by $\gamma$-approximation. Furthermore, our regret bound closely matches that of nearly minimax-optimal algorithms for MNL bandits under uniform revenues (Lee & Oh, 2024). The difference lies in the dimensionality: in our DMNL setting, the agent must learn both the relevance parameter $\boldsymbol{\theta}^*$ and the diversity parameter $\lambda^*$, whereas existing MNL bandits only require estimation of $\boldsymbol{\theta}^*$. Despite this additional complexity, the matching regret bound implies that the proposed algorithm remains statistically efficient while explicitly accounting for diversity.

Unlike prior works in combinatorial bandits that model diversity through an explicit balance between a submodular diversity function and an additive reward function (Chen et al., 2013; Qin et al., 2014; Chen et al., 2016), our DMNL framework jointly learns both the relevance parameter $\boldsymbol{\theta}^*$ and the diversity parameter $\lambda^*$. As a result, our method does not require manually tuning hyperparameters to balance relevance and diversity. Furthermore, the proposed algorithm leverages item-wise optimistic construction based on the submodularity of the reward function, achieving computational efficiency (with $\mathcal{O}(NK)$ cost per round) and provably improved approximation ratio—without relying on a black-box optimization oracle often assumed in combinatorial bandit literature. From a technical perspective, even though the agent does not receive intermediate feedback on the marginal reward gain for individual items, we show that the strict submodularity of the diversity score function is sufficient to guarantee the submodularity of the overall optimistic reward function $\widetilde{R}_t(S)$. This allows us to maintain provable performance guarantees without requiring marginal gain feedback, which is typically assumed in prior submodular bandit settings (Yue & Guestrin, 2011; Chen et al., 2017; Hiranandani et al., 2020). The detailed proof is provided in Appendix F.

## 4 NUMERICAL EXPERIMENTS

We evaluate the empirical performance of our proposed algorithm against several baselines in the DMNL bandit setting. These include existing MNL bandit algorithms: UCB-MNL (Oh & Iyengar, 2021), TS-MNL (Oh & Iyengar, 2019), and OFU-MNL+ (Lee & Oh, 2024), as well as two additional variants of OFU-MNL+ adapted to incorporate diversity.

First, we consider OFU-MNL-DR (Algorithm 2), which follows the existing MNL choice model but uses a submodular reward function of the form $R'_t(S, \boldsymbol{\theta}^*, \lambda) := \frac{\sum_{j \in S} \exp(\mathbf{x}_{tj}^\top \boldsymbol{\theta}^*)}{1 + \sum_{j \in S} \exp(\mathbf{x}_{tj}^\top \boldsymbol{\theta}^*)} + \lambda g(S)$, where $g(S)$ is the diversity score and $\lambda$ is a predefined balancing parameter. Note that OFU-MNL-DR requires tuning $\lambda$ manually, unlike our approach which learns diversity directly.

Second, we include OFU-DMNL-FULL (Algorithm 3), which exactly implements the DMNL model via exhaustive search. It computes $\widetilde{R}_t(S)$ for all $\binom{N}{K}$ subsets at each round, incurring a computational cost of roughly $\mathcal{O}\left( \left( \frac{eN}{K} \right)^K \right)$ per round. These two variants help illustrate the benefit of learning

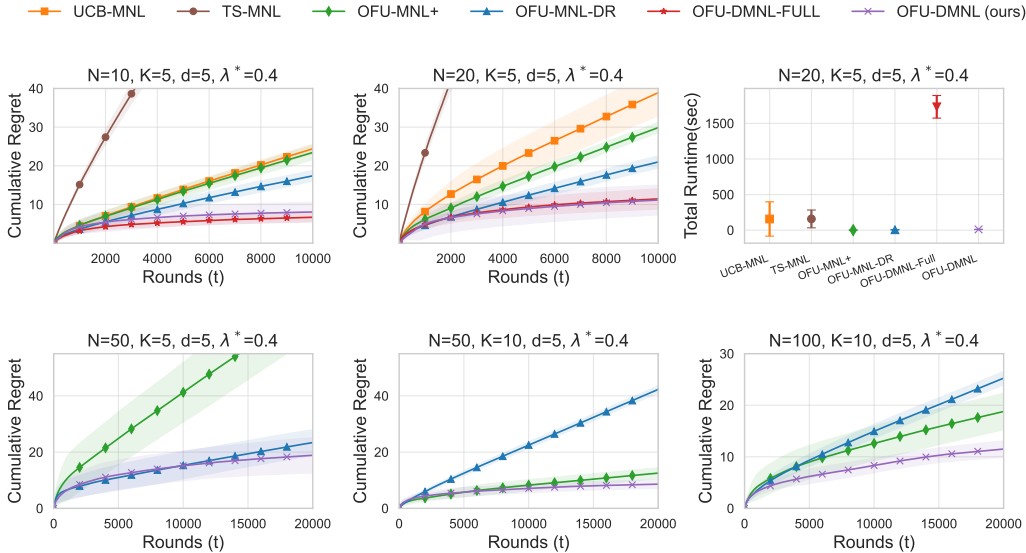

Figure 2: Performance comparison between algorithms. The top row shows cumulative regret (left two, $N = 10, 20$) and total runtime (rightmost, $T = 10000$), and the bottom row shows the cumulative regret of the top 3 algorithms under various parameter settings.

diversity directly (vs. tuning it manually) and the computational trade-off of our efficient item-wise optimistic construction relative to exhaustive search. Details on the implementation of these two variants are provided in Appendix G.1.

For each round, the context features are independently drawn from a Gaussian distribution $\mathcal{N}(\mathbf{0}_d, \mathbf{I}_d)$ and clipped to the range $[-1/\sqrt{d}, , 1/\sqrt{d}]^d$. Each item is also assigned a category, ans the diversity function on an assortment $S$ is then defined as the exponential decaying categorical function (Example 1). To assess how effectively the proposed algorithm adapts to relevance–diversity trade-offs, we fix the diversity parameter $\lambda^*$ at several values. We then sample the relevance parameter $\boldsymbol{\theta}^*$ from a uniform distribution over $[-1/\sqrt{d}, 1/\sqrt{d}]^d$ and scale it to satisfy $\|\boldsymbol{\theta}^*\|_2 + \lambda^* = 1$. We conducted 10 independent runs for each configuration,, and all reported results are averaged over these runs.

As shown in Figure 2, our algorithm exhibits superior performance compared to the baseline algorithms. Notably, it achieves competitive regret performance relative to the exhaustive-search algorithm, `OFU-DMNL-FULL`, while demonstrating a dramatic advantage in runtime efficiency. Moreover, our proposed algorithm outperforms both `OFU-MNL+` and `OFU-MNL-DR` across various problem sizes and under various configurations that control the balance between relevance and diversity (controlled by $\lambda$, refer to Figure 5). These results highlight the robustness of our method in handling different trade-off regimes between item relevance and assortment diversity. Detailed experimental settings and additional results under various configurations are provided in Appendix G.

## 5  CONCLUSION

In this paper, we propose the diversified multinomial logit contextual bandit, a new model that captures the trade-off between item relevance and assortment diversity. To solve this problem, we design a UCB-based algorithm that incrementally constructs assortment via item-wise optimistic utility estimates. Unlike prior works relying on black-box optimization oracles, our approach employs a white-box, item-wise construction strategy with a provable approximation guarantee of at least $(1 - \frac{1}{e+1})$. We further show that the algorithm achieves a $(1 - \frac{1}{e+1})$-approximate cumulative regret bound of $\widetilde{\mathcal{O}}(d\sqrt{T/K})$, matching the nearly minimax regret of MNL bandits despite the added challenge of jointly learning a diversity parameter—highlighting both statistical efficiency and modeling generality. Empirical results demonstrate superior performance across a wide range of scenarios with significantly lower computational cost. Overall, our work offers a practical and theoretically grounded solution for diversity-aware sequential decision-making.

## ACKNOWLEDGEMENTS

This work was supported by the National Research Foundation of Korea (NRF) grant and the Institute of Information & communications Technology Planning & Evaluation (IITP) grant both funded by the Korea government (MSIT) (No. RS-2022-NR071853, RS-2023-00222663, RS-2025-25421879, RS-2025-25463302) and by AI-Bio Research Grant through Seoul National University.

## LLM USAGE

We employed an LLM for typo correction and grammar checking.

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

CONTENTS OF APPENDIX

# A   DEFINITIONS AND NOTATIONS

Recall that we define $N$ as the total number of items and $\mathcal{S}$ as the set of candidate assortments with a size constraint of at most $K$, i.e., $\mathcal{S} = \{S \subset [N] : |S| \leq K\}$.

**Definition 3** (Monotone increasing set function). *The set function $f$ mapping sets $S \in \mathcal{S}$ to a real-valued number is monotone if and only if for every $S, S' \in \mathcal{S}$ with $S \subseteq S'$, $f$ satisfies $f(S) \leq f(S')$,*

**Definition 4** (Submodular function). *The set function $f$ mapping sets $S \in \mathcal{S}$ to a real-valued number is submodular if and only if for every $S \subseteq S'$ and $e \notin S'$, $f$ satisfies*

$$f(S' \cup \{e\}) - f(S') \leq f(S \cup \{e\}) - f(S).$$

*Or, equivalently, for $e_1, e_2 \notin S$, we have*

$$f(S \cup \{e_1\}) - f(S) \geq f(S \cup \{e_1, e_2\}) - f(S \cup \{e_2\}).$$

For convenience, we provide a table summarizing the notations.

Table 1: Notations

| | |
|---|---|
| $N$ | total number of items |
| $K$ | maximum size of assortmens |
| $d$ | dimension of feature vectors |
| $T$ | number of total rounds |
| $g_t$ | diversity score function in round $t$ |
| $\mathbf{x}_{ti}$ | feature vector for item $i$ in round $t$ |
| $\mathbf{z}_{ti}(S)$ | $:= [\mathbf{x}_{ti}, g_t(S)]$, diversity-augmented feature vector for item $i$ in round $t$ |
| $0$ | outside option in MNL choice model |
| $\kappa$ | $:= \min_{t \in [T], S \in \mathcal{S}, i \in S, \|\boldsymbol{\theta}\|_2 \leq 1, 0 \leq \lambda \leq 1} p_t(i\|S, \boldsymbol{\theta}, \lambda) p_t(0\|S, \boldsymbol{\theta}, \lambda) > 0$ |
| $S_t$ | selected assortment in round $t$ |
| $S_t^*$ | optimal assortment in round $t$ |
| $y_{ti}$ | user's choice for item $i \in S_t \cup \{0\}$ in round $t$ |
| $R_t(S, \boldsymbol{\theta}^*, \lambda^*)$ | $:= \sum_{i \in S} p_t(i \mid S, \boldsymbol{\theta}^*, \lambda^*)$, reward of the assortment $S$ in round $t$ |
| $\ell_t(\mathbf{w})$ | $:= -\sum_{i \in S_t} y_{ti} \log p_t(i \mid S_t, \mathbf{w})$, loss function in round $t$ |
| $\mathcal{G}_t(\mathbf{w})$ | $:= \nabla^2 \ell_t(\mathbf{w})$ |
| | $= \sum_{i \in S_t} p_t(i \mid S_t, \mathbf{w}) \mathbf{z}_{ti}(S_t) \mathbf{z}_{ti}(S_t)^\top - \sum_{i \in S_t} \sum_{j \in S_t} p_t(i \mid S_t, \mathbf{w}) p_t(j \mid S_t, \mathbf{w}) \mathbf{z}_{ti}(S_t) \mathbf{z}_{tj}(S_t)^\top$ |
| $\Lambda$ | regularization parameter |
| $\mathbf{H}_t$ | $:= \Lambda \mathbf{I}_{d+1} + \sum_{s=1}^{t-1} \mathcal{G}_s(\mathbf{w}_{s+1})$ |
| $\widetilde{\mathbf{H}}_t$ | $:= \mathbf{H}_t + \eta \, \mathcal{G}_t(\mathbf{w}_t)$ |
| $V_t$ | $:= \Lambda \mathbf{I}_{d+1} + \sum_{s=1}^{t} \sum_{i \in S_s} \mathbf{z}_{si}(S_s) \mathbf{z}_{si}(S_s)^\top$, gram matrix |
| $\alpha_t$ | confidence radius |
| $\mathrm{ucb}(\mathbf{z}_{ti}(S))$ | $:= \mathbf{z}_{ti}(S)^\top \mathbf{w}_t + \alpha_t \|\mathbf{z}_{ti}(S)\|_{\mathbf{H}_t^{-1}}$ . |
| $\widetilde{R}_t(S)$ | $:= \sum_{i \in S} \frac{\exp(\mathrm{ucb}(\mathbf{z}_{ti}(S)))}{1 + \sum_{j \in S} \exp(\mathrm{ucb}(\mathbf{z}_{tj}(S)))}$ |
| $f_t(S)$ | $:= \log\left(\sum_{i \in S} \exp(\mathbf{x}_{ti}^\top \boldsymbol{\theta}^* + \lambda^* g_t(S))\right)$ |
| $\widetilde{f}_t(S)$ | $:= \log\left(\sum_{i \in S} \exp\left(\mathbf{z}_{ti}(S)^\top \mathbf{w}_t + \alpha_t \|\mathbf{z}_{ti}(S)\|_{\mathbf{H}_t^{-1}}\right)\right)$ . |

# B   RELATED WORK

**Contextual MNL bandits.**   The MNL bandit framework—which applies the MNL choice model to dynamic assortment optimization—has led to significant progress in sequential decision-making (Rusmevichientong et al., 2010; Sauré & Zeevi, 2013; Agrawal et al., 2017; Cheung & Simchi-Levi, 2017; Ou et al., 2018; Agrawal et al., 2019; Oh & Iyengar, 2019; Chen et al., 2020; Oh & Iyengar, 2021; Perivier & Goyal, 2022; Lee & Oh, 2024; Zhang & Sugiyama, 2024; Lee & Oh, 2025). Early results established the statistical efficiency of UCB-type (Agrawal et al., 2017)

and Thompson Sampling (TS)-type (Agrawal et al., 2019) algorithms, while Chen & Wang (2017) derived lower bounds for the MNL bandit problem. The contextual MNL bandit was introduced by Oh & Iyengar (2019), who proposed a TS-type algorithm with provable guarantees, followed by a UCB-type algorithm (Oh & Iyengar, 2021) achieving $\widetilde{\mathcal{O}}(\sqrt{dT/\kappa})$ regret, where $\kappa = \mathcal{O}(1/K^2)$ is an instance-dependent parameter. Subsequent works refined the theoretical analysis: Perivier & Goyal (2022) derived tighter regret bounds, and Lee & Oh (2024; 2025) proposed computationally efficient algorithms with nearly minimax guarantees. However, despite these advances, no prior work has incorporated diversity into the MNL bandit model to better reflect user preferences for varied assortments.

**Submodular bandits and combinatorial bandits.** Handling the computational problem via the submodular reward function in bandit setting is first explored by Yue & Guestrin (2011). They introduced the linear submodular bandit framework, in which the reward of a set of items is assumed by a linear combination of submodular set functions. Built on the fact that by using submodular reward functions, item-wise selection (iteratively adding one item at a time in a greedy manner) for set construction guarantees the approximation rate $(1 - \frac{1}{e})$ for submodular rewards (Nemhauser et al., 1978), they suggest an algorithm that item-wisely selects items during set optimization and proved a theoretical bound for their algorithm with $(1 - \frac{1}{e})$-approximate regret.

In the submodular bandit framework (Yue & Guestrin, 2011; Chen et al., 2017; Hiranandani et al., 2020), the set of items is either ranked or presented sequentially to the user, and the probability of an item being selected depends on its marginal gain relative to the previously presented items. Especially, Hiranandani et al. (2020) proposed a cascade variant of the model suggested by Yue & Guestrin (2011). It is notable that these settings are fundamentally different from the assortment bandit problem we study, since the feedback is determined by a user choice at the end. While we aim to exploit submodularity to design computationally tractable algorithms, unlike in submodular bandits, we cannot leverage information about the marginal gain obtained when items are added individually. Therefore, whether the optimization advantages of submodular diversity functions can be applied to the MNL framework remains an open direction.

In the combinatorial bandit framework (Chen et al., 2013; Qin et al., 2014; Chen et al., 2016; Li et al., 2016; Hwang et al., 2023), the reward of a set of items is defined as a function of the rewards of the individual arms, which allows optimization to exploit properties of the set such as diversity. However, the key difference from the assortment bandit is that the expected reward of each individual item is unaffected by the other items in the set; consequently, the properties of the set influence only the reward, not the choice model. In particular, when modeling diversity within the combinatorial bandit framework, the diversity parameter must be given in advance as a hyperparameter. This is fundamentally different from our setting, in which diversity is embedded into the MNL choice probability model and the algorithm must estimate the corresponding parameters.

## C STRICT SUBMODULARITY

### C.1 CHALLENGES IN ITEM-WISE OPTIMISTIC CONSTRUCTION

**Lack of intermediate feedback.** In the item-wise optimistic construction process in Eq.(6), when selecting the $k$-th item $a_{t,k}$ (for $k \in [K]$), the current partial assortment consists of the previously selected items $a_{t,1}, \ldots, a_{t,k-1}$ (Line 9 in Algorithm 1). Therefore, the optimistic diversity-augmented utility in Eq.(4) is computed using only the diversity score of the partial assortment, $g(\{a_{t,1}, \ldots, a_{t,k-1}\})$. However, the agent does not receive any feedback about the diversity score at the time of item addition, nor does it observe the marginal gain in reward for each individual item—even after offering the full assortment. Instead, it only observes the total reward associated with the final constructed set $S_t = \{a_{t,1}, \ldots, a_{t,K}\}$. This lack of intermediate feedback complicates the analysis of regret, particularly in submodular bandit settings that rely on item-wise optimistic construction.

On the other hand, prior works on submodular bandits (Yue & Guestrin, 2011; Chen et al., 2017; Hiranandani et al., 2020) assume access to intermediate feedback on the marginal gain of each item during or after the construction of an assortment. For example, Yue & Guestrin (2011) receives slot-level feedback after offering an assortment (i.e., a list of articles), which enables the agent to

estimate the marginal utility contribution of each item. Similarly, in the cascading bandit model of Hiranandani et al. (2020), items are presented in a ranked list and examined sequentially by the user. This naturally reveals the marginal utility of each item conditioned on the items already shown. In Chen et al. (2017), the agent receives explicit "interactive feedback" on the marginal gain of each newly added item during the construction process, making the feedback even more granular. We also emphasize that this challenge does not arise in the combinatorial bandit literature (Chen et al., 2013; Qin et al., 2014; Chen et al., 2018; Hwang et al., 2023), where the diversity of selected arms is incorporated solely through the reward function. Since diversity is not parameterized in those models, there is no need to estimate any diversity-related parameters during learning.

**Beyond submodular diversity.** One way to overcome the intermediate feedback problem is to exploit the submodularity of the diversified optimistic expectd reward in Eq.(5). For an assortment $S$, let us define $\widetilde{f}_t(S)$ as follows:

$$\widetilde{f}_t(S) := \log \left( \sum_{i \in S} \exp \left( \mathbf{z}_{ti}(S)^\top \mathbf{w}_t + \alpha_t \|\mathbf{z}_{ti}(S)\|_{\mathbf{H}_t^{-1}} \right) \right). \tag{8}$$

We note that if $\widetilde{f}_t(S)$ is submodular, then we show that the assortment $S_t$ constructed by the item-wise optimistic construction can approximate the true optimal assortment of $\widetilde{f}_t$, making it possible to establish bounds without relying on feedback on marginal gain. However, unfortunately, while the submodularity of $g_t$ guarantees that $f_t$ is submodular (Section 3.1), it does not ensure the submodularity of $\widetilde{f}_t$.

To check whether $\widetilde{f}_t$ is submodular or not, for any $S \in \mathcal{S}$, and $e_1, e_2 \notin S$, define $S_1 := S \cup \{e_1\}$, $S_2 := S \cup \{e_2\}$, $S_3 := S \cup \{e_1, e_2\}$. Then,

$$\widetilde{f}_t(S_1) - \widetilde{f}_t(S) - \left( \widetilde{f}_t(S_3) - \widetilde{f}_t(S_2) \right)$$

$$= \log \left( \frac{\sum_{i \in S_1} \exp \left( \mathbf{x}_{ti}^\top \hat{\boldsymbol{\theta}}_t + \hat{\lambda}_t g_t(S_1) + \alpha_t \|[\mathbf{x}_{ti}, g_t(S_1)]\|_{\mathbf{H}_t^{-1}} \right)}{\sum_{i \in S} \exp \left( \mathbf{x}_{ti}^\top \hat{\boldsymbol{\theta}}_t + \hat{\lambda}_t g_t(S) + \alpha_t \|[\mathbf{x}_{ti}, g_t(S)]\|_{\mathbf{H}_t^{-1}} \right)} \right.$$

$$\left. \times \frac{\sum_{i \in S_2} \exp \left( \mathbf{x}_{ti}^\top \hat{\boldsymbol{\theta}}_t + \hat{\lambda}_t g_t(S_2) + \alpha_t \|[\mathbf{x}_{ti}, g_t(S_2)]\|_{\mathbf{H}_t^{-1}} \right)}{\sum_{i \in S_3} \exp \left( \mathbf{x}_{ti}^\top \hat{\boldsymbol{\theta}}_t + \hat{\lambda}_t g_t(S_3) + \alpha_t \|[\mathbf{x}_{ti}, g_t(S_3)]\|_{\mathbf{H}_t^{-1}} \right)} \right)$$

$$= \log \left( \frac{\sum\limits_{i \in S, j \in S_3} \exp \left( \mathbf{x}_{ti}^\top \hat{\boldsymbol{\theta}}_t + \mathbf{x}_{tj}^\top \hat{\boldsymbol{\theta}}_t + \hat{\lambda}_t g_t(S_1) + \hat{\lambda}_t g_t(S_2) + \alpha_t \left( \|[\mathbf{x}_{ti}, g_t(S_1)]\|_{\mathbf{H}_t^{-1}} + \|[\mathbf{x}_{tj}, g_t(S_2)]\|_{\mathbf{H}_t^{-1}} \right) \right)}{\sum\limits_{i \in S, j \in S_3} \exp \left( \mathbf{x}_{ti}^\top \hat{\boldsymbol{\theta}}_t + \mathbf{x}_{tj}^\top \hat{\boldsymbol{\theta}}_t + \hat{\lambda}_t g_t(S) + \hat{\lambda}_t g_t(S_3) + \alpha_t \left( \|[\mathbf{x}_{ti}, g_t(S)]\|_{\mathbf{H}_t^{-1}} + \|[\mathbf{x}_{tj}, g_t(S_3)]\|_{\mathbf{H}_t^{-1}} \right) \right)} \right.$$

$$\left. + \frac{\exp \left( \mathbf{x}_{t,e_1}^\top \hat{\boldsymbol{\theta}}_t + \mathbf{x}_{t,e_2}^\top \hat{\boldsymbol{\theta}}_t + \hat{\lambda}_t g_t(S_1) + \hat{\lambda}_t g_t(S_2) + \alpha_t \left( \|[\mathbf{x}_{t,e_1}, g_t(S_1)]\|_{\mathbf{H}_t^{-1}} + \|[\mathbf{x}_{t,e_2}, g_t(S_2)]\|_{\mathbf{H}_t^{-1}} \right) \right)}{\sum\limits_{i \in S, j \in S_3} \exp \left( \mathbf{x}_{ti}^\top \hat{\boldsymbol{\theta}}_t + \mathbf{x}_{tj}^\top \hat{\boldsymbol{\theta}}_t + \hat{\lambda}_t g_t(S) + \hat{\lambda}_t g_t(S_3) + \alpha_t \left( \|[\mathbf{x}_{ti}, g_t(S)]\|_{\mathbf{H}_t^{-1}} + \|[\mathbf{x}_{tj}, g_t(S_3)]\|_{\mathbf{H}_t^{-1}} \right) \right)} \right).$$

As in prior approaches to ensure the submodularity of the LogSumExp function, one would need to show that the numerator of the first term inside the logarithm is larger than its denominator. However, this does not hold in general. In general, even though $g$ is submodular, the following inequality does not hold:

$$\|[\mathbf{x}_{j_1}, g_t(S_1)]\|_{\mathbf{H}_t^{-1}} + \|[\mathbf{x}_{j_2}, g_t(S_2)]\|_{\mathbf{H}_t^{-1}} \geq \|[\mathbf{x}_{j_1}, g_t(S)]\|_{\mathbf{H}_t^{-1}} + \|[x_{j_2}, g_t(S_3)]\|_{\mathbf{H}_t^{-1}}. \tag{9}$$

Specifically when $j_1 = j_2 = j$, we can prove that the left hand side is negative, since the weighted norm has convex structure. This means that the item-wise optimistic construction cannot, in general, guarantee $(1 - \frac{1}{e})$-approximate optimality with respect to $\widetilde{f}_t$. However, we show that the submodularity of $\widetilde{f}_t$ can be recovered if the diversity function $g_t$ satisfies $\omega$-strict submodularity, as established in Lemma 3.

## C.2 Examples of Strict Submodular Diversity Functions

In this section, we present several examples of strictly submodular set functions that are applicable to a wide range of practical settings.

**Example 1** (Categorical functions). *For a given item set $S$, let $n_S$ denote the number of categories that can be covered by $S$, i.e., the size of the set of categories to which the items in $S$ belong.*

- *(i) **Exponential decaying case.** Consider the case when an item from the $m$-th new category is added, the value of $g_\rho(S)$ increases by $\rho^{m-1}$ for $\rho \in (0,1)$. Then, the diversity function is defined by $g_\rho(S) := 1 + \rho + \rho^2 + \cdots + \rho^{n_S-1}$ and is $(1-\rho)$-strict submodular.*

- *(ii) **Polynomial decaying case.** If the diversity function is defined by $g_\alpha(S) := (n_S)^\alpha$ for $\alpha \in (0,1)$, then, $g_\alpha(S)$ is $\left(1 - \left(\frac{2M}{2M+1}\right)^{1-\alpha}\right)$-strict submodular, where $M$ is the maximum number of categories.*

*Proof of Example 1.* We define $M$ as the number of categories, $c(i) \in [M]$ the category that the item $i \in [N]$ belongs to, and $c(S) := \{c(i) \mid i \in S\}$ ($|c(S)| = n_s$). Let $S' = S \cup \{e'\}$, $e \notin S'$. To show $\omega$-strict submodularity of the set function $g$, it is enough to show that if $g(S) \neq g(S')$, then the following holds.

$$g(S' \cup \{e\}) - g(S') \leq (1-\omega)[g(S \cup \{e\}) - g(S)] \qquad (10)$$

We first show that for any $\rho \in (0,1)$, the exponential decaying categorical function $g_\rho$ is $(1-\rho)$-strict submodular. Suppose that $S$ and $S'$ satisfy $g_\rho(S) \neq g_\rho(S')$. Then, by the definition of $g_\rho$, $c(e') \notin c(S)$. Thus,

$$g_\rho(S) = 1 + \rho + \ldots + \rho^{n_S-1}$$
$$g_\rho(S') = 1 + \rho + \ldots + \rho^{n_S}.$$

If $c(e) \in c(S)$, then $g_\rho(S \cup \{e\}) - g_\rho(S) = 0 = g_\rho(S' \cup \{e\}) - g_\rho(S')$. Else if $c(e) \in c(S')\backslash c(S)$, i.e. $c(e) = c(e')$, then $g_\rho(S \cup \{e\}) - g_\rho(S) = \rho^{n_S}$ and $g_\rho(S' \cup \{e\}) - g_\rho(S') = 0$, and hence the inequality (*) holds for all $\omega \in (0,1)$. Otherwise, if $c(e) \notin c(S')$, then

$$g_\rho(S \cup \{e\}) - g_\rho(S) = \rho^{n_S}$$
$$g_\rho(S' \cup \{e\}) - g_\rho(S') = \rho^{n_S+1},$$

and so, $g_\rho(S' \cup \{e\}) - g_\rho(S') = \rho[g_\rho(S \cup \{e\}) - g_\rho(S)]$ holds for $\omega = 1 - \rho$.

For all cases, the function $g_\rho$ satisfies condition in Eq.(10) for $\omega = 1 - \rho$, and therefore it is $(1-\rho)$-strict submodular.

Secondly, we will show that for any $\alpha \in (0,1)$, the polynomial decaying categorical function $g_\alpha$ is $\left(1 - \left(\frac{2M}{2M+1}\right)^{1-\alpha}\right)$-strict submodular. By the same reasoning as in the exponential decaying case, it suffices that condition (*) holds for $\omega = \left(1 - \frac{2M}{2M+1}\right)^{1-\alpha}$ only when $c(e') \in c(S')\backslash c(S)$ and $c(e) \notin c(S')$. If $c(e') \in c(S')\backslash c(S)$ and $c(e) \notin c(S')$, then

$$g_\alpha(S \cup \{e\}) - g_\alpha(S) = (n_S + 1)^\alpha - (n_S)^\alpha$$
$$g_\alpha(S' \cup \{e\}) - g_\alpha(S') = (n_S + 2)^\alpha - (n_S + 1)^\alpha.$$

Let $h(x) = x^\alpha$ for $\alpha \in (0,1)$. Since $h$ is a increasing concave function, $h'(x+1) < h(x+1) - h(x) < h'(x + \frac{1}{2})$ holds. Thus, $(n_S+1)^\alpha - (n_S)^\alpha > \frac{\alpha}{(n_S+1)^{1-\alpha}}$ and $(n_S+2)^\alpha - (n_S+1)^\alpha < \frac{\alpha}{(n_S+\frac{3}{2})^{1-\alpha}}$ hold, and hence,

$$\frac{g_\alpha(S' \cup \{e\}) - g_\alpha(S')}{g_\alpha(S \cup \{e\}) - g_\alpha(S)} = \frac{(n_S+2)^\alpha - (n_S+1)^\alpha}{(n_S+1)^\alpha - (n_S)^\alpha}$$

$$< \frac{(n_S+1)^{1-\alpha}}{(n_S+\frac{3}{2})^{1-\alpha}} = \frac{1}{\left(1 + \frac{1}{2(n_S+1)}\right)^{1-\alpha}}$$

$$\leq \left(\frac{2M}{2M+1}\right)^{1-\alpha}.$$

Therefore, the function $g_\alpha$ satisfies condition in Eq.(10) for $\omega = 1 - \left(\frac{2M}{2M+1}\right)^{1-\alpha}$, and therefore it is $(1 - \left(\frac{2M}{2M+1}\right)^{1-\alpha})$-strict submodular. $\qquad\square$

**Example 2** (Categorical level functions)**.** *For each item $i \in [N]$, let $c(i) \in \mathbb{R}^M$ be the categorical feature vector of item $i$. Each category $m \in [M]$ has $n_m$ levels, and the $m$-th entry of $c(i)$ has a value of $c_m(i) \in \{0, \frac{1}{n_m}, \frac{2}{n_m}, \ldots, \frac{n_m}{n_m} = 1\}$. Define $c(S) := \sum_{m \in [M]} \max_{i \in S} c_m(i)$ be a sum of categorical level-coverage of items in $S$, and for $\rho \in (0, 1)$ and $k > 0$, $g_{\rho,k}(S) = k\left(1 - \rho^{c(S)}\right)$ be a categorical function on $S \in \mathcal{S}$.*

**Proposition 1.** *For any $\rho \in (0, 1)$ and $k > 0$, $g_{\rho,k}$ is $(1 - \rho^\Delta)$-strict submodular, where $\Delta := \min_{m \in [M]} \frac{1}{n_m} = \frac{1}{\max_{m \in [M]} n_m}$.*

*Proof of Proposition 1.* Let $S' = S \cup \{e'\}$ and $e \notin S'$. To show strict submodularity, it is enough to show that if $g_{\rho,k}(S) \neq g_{\rho,k}(S')$, then the following holds.

$$g_{\rho,k}(S' \cup \{e\}) - g_{\rho,k}(S') \leq \rho^\Delta [g_{\rho,k}(S \cup \{e\}) - g_{\rho,k}(S)] \tag{11}$$

Suppose that $S$ and $S'$ satisfy $g_{\rho,k}(S) \neq g_{\rho,k}(S')$. Since $h(x) = k(1 - \rho^x)$ is a one-to-one function, we have $c(S) \neq c(S')$. Then, by the definition of $\Delta$, it holds that $c(S') - c(S) \geq \Delta$.

By the (level-coverage) definition of the function $c$, $c$ is submodular, and hence,

$$c(S' \cup \{e'\}) - c(S') \leq c(S \cup \{e\}) - c(S). \tag{12}$$

Then,

$$
\begin{aligned}
g(S' \cup \{e\}) - g(S') &= k\left(1 - \rho^{c(S' \cup \{e\})}\right) - k\left(1 - \rho^{c(S')}\right) \\
&= k\left(\rho^{c(S')} - \rho^{c(S' \cup \{e\})}\right) \\
&\leq k\left(\rho^{c(S')} - \rho^{c(S \cup \{e\}) + c(S') - c(S)}\right) \qquad (\because \rho < 1 \text{ and Eq. 12}) \\
&\leq k\rho^{c(S')}\left(1 - \rho^{c(S \cup \{e\}) - c(S)}\right) \\
&\leq k\rho^{c(S) + \Delta}\left(1 - \rho^{c(S \cup \{e\}) - c(S)}\right) \qquad (\because \rho < 1 \text{ and } c(S') \geq c(S) + \Delta) \\
&= k\rho^\Delta\left(\rho^{c(S)} - \rho^{c(S \cup \{e\})}\right) \\
&= k\rho^\Delta\left[\left(1 - \rho^{c(S \cup \{e\})}\right) - \left(1 - \rho^{c(S)}\right)\right] \\
&= \rho^\Delta[g(S' \cup \{e\}) - g(S')],
\end{aligned}
$$

which results in that $g$ is $(1 - \rho^\Delta)$-strict submodular. $\qquad\square$

**Remark 4.** *In Example 2, we simply use $c(S) := \sum_{m \in [M]} \max_{i \in S} c_m(i)$ and $g_{\rho,k}(S) = h(c(S))$ where $h(x) := k(1 - \rho^x)$. However, if $c$ is of a form with minimum increase (e.g., max or sum) and $h$ is chosen as a strictly concave function, then $g = h(c(S))$ becomes $\omega$-strict submodular for some $\omega \in (0, 1)$.*

## D    PROOF OF THEOREM 1

*Proof of Theorem 1.* For an assortment $S \in \mathcal{S}$, let us define $f_t(S)$ as follows:

$$f_t(S) = \log\left(\sum_{j \in S} \exp(\mathbf{x}_{tj}^\top \boldsymbol{\theta}^* + \lambda^* g_t(S))\right) = \log\left(\sum_{j \in S} \exp(\mathbf{x}_{tj}^\top \boldsymbol{\theta}^*)\right) + \lambda^* g_t(S). \tag{13}$$

Also, we abbreviate $R_t(S, \boldsymbol{\theta}^*, \lambda^*)$ as $R_t(S)$. If we let $\psi_t := \exp(f_t(S_t^{\text{greedy}}))$, then, by the definition of $f_t$,

$$R_t(S_t^{\text{greedy}}) = \frac{\psi_t}{1 + \psi_t}.$$

By the additivity of submodular functions, $f_t$ is monotone and submodular, and hence the greedy solution for maximizing $f_t$ can achieve $(1 - \frac{1}{e})$-approximation rate (Nemhauser et al., 1978), which means

$$f_t(S_t^{\text{greedy}}) \geq \left(1 - \frac{1}{e}\right) f_t(S_t^*).$$

Therefore, we have

$$R_t(S_t^*) = \frac{\exp(f_t(S_t^*))}{1 + \exp(f_t(S_t^*))} \leq \frac{\exp\left(\frac{e}{e-1} \cdot f_t(S_t^{\text{greedy}})\right)}{1 + \exp\left(\frac{e}{e-1} \cdot f_t(S_t^{\text{greedy}})\right)} = \frac{\psi_t^\alpha}{1 + \psi_t^\alpha},$$

where we denote $\alpha = \frac{e}{e-1}$.

To get the approximation rate, we want to bound the below function

$$h(\psi) := \left(\frac{\psi^\alpha}{1 + \psi^\alpha}\right) / \left(\frac{\psi}{1 + \psi}\right) = \frac{\psi^\alpha(1 + \psi)}{\psi(1 + \psi^\alpha)}.$$

Since $h'(\psi) = \frac{1}{(\psi + \psi^{\alpha+1})^2} \times \psi^\alpha \left(-\psi^\alpha + \alpha\psi + (\alpha - 1)\right)$, the equation $h'(\psi) = 0$ has a unique solution $\psi_0 > 0$, which is the maximum point in $\mathbb{R}_+$. Since $h$ has the maximum at $\psi_0$ in $\mathbb{R}_+$,

$$\frac{R_t(S_t^*)}{R_t(S_t^{\text{greedy}})} \leq \left(\frac{\psi_t^\alpha}{1 + \psi_t^\alpha}\right) / \left(\frac{\psi_t}{1 + \psi_t}\right) = h(\psi_t) \leq h(\psi_0),$$

which implies that

$$R_t(S_t^{\text{greedy}}) \geq \frac{1}{h(\psi_0)} R_t(S_t^*) = \frac{\psi_0(1 + \psi_0^\alpha)}{\psi_0^\alpha(1 + \psi_0)} R_t(S_t^*).$$

Since $\psi_0$ satisfies $\psi_0^\alpha = \alpha\psi_0 + (\alpha - 1)$, we have $\psi_0 > 1$, and hence,

$$
\begin{aligned}
h(\psi_0) &= \frac{\psi_0^\alpha + \psi_0^{\alpha+1}}{\psi_0 + \psi_0^{\alpha+1}} = \frac{\alpha\psi_0 + (\alpha - 1) + \alpha\psi_0^2 + (\alpha - 1)\psi_0}{\psi_0 + \alpha\psi_0^2 + (\alpha - 1)\psi_0} \\
&= \frac{\alpha\psi_0^2 + (2\alpha - 1)\psi_0 + (\alpha - 1)}{\alpha\psi_0^2 + \alpha\psi_0} = \frac{(\alpha - 1)(\psi_0 + 1)}{\alpha\psi_0 + \alpha\psi_0} = 1 + \frac{\alpha - 1}{\alpha\psi_0} \\
&< 1 + \frac{1}{e}.
\end{aligned}
$$

Therefore, the approximation rate $\frac{1}{h(\psi_0)}$ is greater than $\frac{e}{e+1}$, which results in

$$\left(1 - \frac{1}{e+1}\right) R_t(S_t^*) \leq \frac{1}{h(\psi_0)} R_t(S_t^*) \leq R_t(S_t^{\text{greedy}}).$$

$\square$

# E  LOWER BOUND

## E.1  PROOF OF THEOREM 2

The proof closely follows the lower-bound arguments developed for the MNL bandit setting (Chen et al., 2020; Lee & Oh, 2024). However, unlike the standard MNL setting—where the diversity function $g$ can be treated as a constant—our framework imposes Assumption 2, which prevents $g$ from being a constant function. Consequently, we derive a lower bound where $g$ is non-constant and strict submodular.

*Proof of Theorem 2.* Let $\lambda = \frac{1}{2}$ and $\epsilon \in \left(0, 1/d^{3/2}\right)$ that will be specified later. For every subset $V \subset [d]$, we define $\boldsymbol{\theta}_V \in \mathbb{R}^d$ as $[\boldsymbol{\theta}_V]_j = \epsilon$ for $j \in V$, and $[\boldsymbol{\theta}_V]_j = 0$ for $j \notin V$, and $\Theta := \{\boldsymbol{\theta}_V : V \subset \mathcal{V}_{d/4}\}$ where $\mathcal{V}_{d/4} = \{V \subset [d], |V| = \frac{d}{4}\}$. Then, for $V \in \mathcal{V}_{d/4}$, $\|\boldsymbol{\theta}_V\|_2 \leq \sqrt{\frac{d\epsilon^2}{4}} \leq \frac{1}{2}$.

We consider the $K \times |\mathcal{V}_{d/4}|$ context vectors invariant across rounds $t$. For each $U \in \mathcal{V}_{d/4}$, there are identical $K$ context vectors with feature $x_U$, where $[x_U]_j = 1/\sqrt{d}$ for $j \in U$ and $[x_U]_j = 0$ for $j \notin U$. Then, for $U \in \mathcal{V}_{d/4}$, $\|x_U\|_2 \leq \sqrt{\frac{d}{4} \cdot \frac{1}{d}} = \frac{1}{2}$.

Let $U_0 = [d/4] \in \mathcal{V}_{d/4}$ and $x_0 := x_{U^0} = \left(\frac{1}{\sqrt{d}}, \frac{1}{\sqrt{d}}, \ldots, \frac{1}{\sqrt{d}}, 0, \ldots, 0\right)$. We define $g(S) := 1$ only if there exists $U \neq U_0$ such that $x_U \in S$, and otherwise (i.e. $S$ contains only $x_0$'s), we set $g(S) := 0$. Then $g$ satisfies $0 \leq g(S) \leq 1$ (Assumption 1). Since $g(S_0) = 0$ for $S_0 = \{x_0, \ldots, x_0\}$ and $g(S) = 1$ for all $S \neq S_0$ with size $k$, $g$ satisfies Assumption 2. Furthermore, $g$ is monotone and strict submodular for all $\omega \in (0, 1)$.

Since the worst-case regret in the worst-case problem instances is bounded below by the average of the worst-case expected regret of parameter instances in $\Theta$, we obtain

$$\sup_{\boldsymbol{\theta}, \lambda} \mathbb{E}^{\pi}_{\boldsymbol{\theta}, \lambda} \left[\mathcal{R}(T|\boldsymbol{\theta}, \lambda)\right] = \sup_{\boldsymbol{\theta}, \lambda} \mathbb{E}^{\pi}_{\boldsymbol{\theta}, \lambda} \left[\sum_{t=1}^{T} R_t(S_t^*, \boldsymbol{\theta}, \lambda) - R_t(S_t, \boldsymbol{\theta}, \lambda)\right]$$

$$\geq \frac{1}{|\mathcal{V}_{d/4}|} \sum_{V \in \mathcal{V}_{d/4}} \mathbb{E}^{\pi}_{\boldsymbol{\theta}, \lambda} \left[\sum_{t=1}^{T} R_t(S_t^*, \boldsymbol{\theta}, \lambda) - R_t(S_t, \boldsymbol{\theta}, \lambda)\right].$$

Let $\{S_t\}_{t=1}^{T}$ be a sequence of assortments generated by $\pi$. For a fixed $V$, we define $\widetilde{S}_t := \{x_{\widetilde{U}_t}, \ldots, x_{\widetilde{U}_t}\}$ as the assortment that contains an identical feature vector $x_{\widetilde{U}_t}$, where $x_{\widetilde{U}_t} := \arg\max_{x_U \in S_t} x_U^{\top} \boldsymbol{\theta}_V$. Furthermore, we simplify notation by $\mathbb{E}_V := \mathbb{E}^{\pi}_{\boldsymbol{\theta}_V, \lambda}$ and $\mathbb{P}_V := \mathbb{P}^{\pi}_{\boldsymbol{\theta}_V, \lambda}$.

By Lemma 1, for any $V \in \mathcal{V}_{d/4}$, we have $\sum_{i \in S^*} p_t(i|S^*, \boldsymbol{\theta}_V, \lambda) - \sum_{i \in S^t} p_t(i|S_t, \boldsymbol{\theta}_V, \lambda) \geq \frac{e^{-1/2}(K-1)}{(e^{-1/2} + Ke)^2} \frac{\delta\epsilon}{2\sqrt{d}}$, where $\delta := d/4 - |\widetilde{U}_t \cap V|$. Thus, we have that

$$\frac{1}{|\mathcal{V}_{d/4}|} \sum_{V \in \mathcal{V}_{d/4}} \mathbb{E}_V \left[\sum_{t=1}^{T} R_t(S_t^*, \boldsymbol{\theta}, \lambda) - R_t(S_t, \boldsymbol{\theta}, \lambda)\right]$$

$$= \frac{1}{|\mathcal{V}_{d/4}|} \sum_{V \in \mathcal{V}_{d/4}} \mathbb{E}_V \left[\sum_{t=1}^{T} \left[\sum_{i \in S^*} p_t(i|S_t, \boldsymbol{\theta}_V, \lambda) - \sum_{i \in S^t} p_t(i|S_t, \boldsymbol{\theta}_V, \lambda)\right]\right]$$

$$\geq \frac{1}{|\mathcal{V}_{d/4}|} \frac{e^{-1/2}(K-1)}{(e^{-1/2} + Ke)^2} \frac{\epsilon}{2\sqrt{d}} \sum_{V \in \mathcal{V}_{d/4}} \mathbb{E}_V \left[\sum_{t=1}^{T} (d/4 - |\widetilde{U}_t \cap V|)\right]$$

$$\geq \frac{e^{-1/2}(K-1)}{(e^{-1/2} + Ke)^2} \frac{\epsilon}{2\sqrt{d}} \left(\frac{dT}{4} - \frac{1}{|\mathcal{V}_{d/4}|} \sum_{V \in \mathcal{V}_{d/4}} \sum_{j \in V} \left[\mathbb{E}_V \left[\sum_{t=1}^{T} \mathbb{1}\{j \in \widetilde{U}_t\}\right]\right]\right)$$

For $j \in V$, we define the random variables $\widetilde{M}_j := \sum_{t=1}^{T} \mathbb{1}\{j \in \widetilde{U}_t\}$. Then, the right hand side is equal to

$$\frac{e^{-1/2}(K-1)}{(e^{-1/2} + Ke)^2} \frac{\epsilon}{2\sqrt{d}} \left(\frac{dT}{4} - \frac{1}{|\mathcal{V}_{d/4}|} \sum_{V \in \mathcal{V}_{d/4}} \sum_{j \in V} \mathbb{E}_V \left[\widetilde{M}_j\right]\right)$$

$$= \frac{e^{-1/2}(K-1)}{(e^{-1/2} + Ke)^2} \frac{\epsilon}{2\sqrt{d}} \left(\frac{dT}{4} - \frac{1}{|\mathcal{V}_{d/4}|} \sum_{V \in \mathcal{V}_{d/4-1}} \sum_{j \notin V} \mathbb{E}_{V \cup \{j\}} \left[\widetilde{M}_j\right]\right)$$

$$\geq \frac{e^{-1/2}(K-1)}{(e^{-1/2} + Ke)^2} \frac{\epsilon}{2\sqrt{d}} \left(\frac{dT}{4} - \frac{|\mathcal{V}_{d/4-1}|}{|\mathcal{V}_{d/4}|} \max_{V \in \mathcal{V}_{d/4-1}} \sum_{j \notin V} \mathbb{E}_{V \cup \{j\}} \left[\widetilde{M}_j\right]\right)$$

$$= \frac{e^{-1/2}(K-1)}{(e^{-1/2}+Ke)^2} \frac{\epsilon}{2\sqrt{d}} \left( \frac{dT}{4} - \frac{|\mathcal{V}_{d/4-1}|}{|\mathcal{V}_{d/4}|} \max_{V \in \mathcal{V}_{d/4-1}} \sum_{j \notin V} \mathbb{E}_V\left[\widetilde{M}_j\right] + \mathbb{E}_{V \cup \{j\}}\left[\widetilde{M}_j\right] - \mathbb{E}_V\left[\widetilde{M}_j\right] \right)$$

$$\geq \frac{e^{-1/2}(K-1)}{(e^{-1/2}+Ke)^2} \frac{\epsilon}{2\sqrt{d}} \left( \frac{dT}{4} - \frac{1}{3} \cdot \frac{dT}{4} - \frac{1}{3} \max_{V \in \mathcal{V}_{d/4-1}} \sum_{j \notin V} \left| \mathbb{E}_{V \cup \{j\}}\left[\widetilde{M}_j\right] - \mathbb{E}_V\left[\widetilde{M}_j\right] \right| \right)$$

$$= \frac{e^{-1/2}(K-1)}{(e^{-1/2}+Ke)^2} \frac{\epsilon}{2\sqrt{d}} \left( \frac{dT}{6} - \frac{1}{3} \max_{V \in \mathcal{V}_{d/4-1}} \sum_{j \notin V} \left| \mathbb{E}_{V \cup \{j\}}\left[\widetilde{M}_j\right] - \mathbb{E}_V\left[\widetilde{M}_j\right] \right| \right)$$

$$\geq \frac{e^{-1/2}(K-1)}{(e^{-1/2}+Ke)^2} \frac{\epsilon}{2\sqrt{d}} \left( \frac{dT}{6} - \frac{1}{3} \max_{V \in \mathcal{V}_{d/4-1}} \sum_{j=1}^{d} \left| \mathbb{E}_{V \cup \{j\}}\left[\widetilde{M}_j\right] - \mathbb{E}_V\left[\widetilde{M}_j\right] \right| \right)$$

$$= \frac{e^{-1/2}(K-1)}{(e^{-1/2}+Ke)^2} \frac{\epsilon}{2\sqrt{d}} \left( \frac{dT}{6} - \frac{1}{3} \sum_{j=1}^{d} \max_{V \in \mathcal{V}_{d/4-1}} \left| \mathbb{E}_{V \cup \{j\}}\left[\widetilde{M}_j\right] - \mathbb{E}_V\left[\widetilde{M}_j\right] \right| \right).$$

We bound $\max_{V \in \mathcal{V}_{d/4-1}} \sum_{j \notin V} \left| \mathbb{E}_{V \cup \{j\}}\left[\widetilde{M}_j\right] - \mathbb{E}_V\left[\widetilde{M}_j\right] \right|$ using KL divergence. By the definition of $\widetilde{M}_j$, we can bound

$$\left| \mathbb{E}_{V \cup \{j\}}\left[\widetilde{M}_j\right] - \mathbb{E}_V\left[\widetilde{M}_j\right] \right| \leq \sum_{t=0}^{T} t \cdot \left| \mathbb{P}_V\left[\widetilde{M}_j = t\right] - \mathbb{P}_{V \cup \{j\}}\left[\widetilde{M}_j = t\right] \right|$$

$$\leq T \cdot \sum_{t=0}^{T} \left| \mathbb{P}_V\left[\widetilde{M}_j = t\right] - \mathbb{P}_{V \cup \{j\}}\left[\widetilde{M}_j = t\right] \right|$$

$$\leq T \cdot \sup_A \left| \mathbb{P}_V(A) - \mathbb{P}_{V \cup \{j\}}(A) \right|$$

$$\leq T \cdot \sqrt{\frac{1}{2}\mathrm{KL}(\mathbb{P}_V \| \mathbb{P}_{V \cup \{j\}})},$$

where the last inequality holds by Pinsker's inequality.

By Lemma 2, we have $\mathrm{KL}(\mathbb{P}_V \| \mathbb{P}_{V \cup \{j\}}) \leq C \cdot \frac{K}{(1+K)^2} \cdot \frac{\mathbb{E}_V\left[\widetilde{M}_j\right]\epsilon^2}{d}$, for some $C > 0$. Therefore,

$$\frac{e^{-1/2}(K-1)}{(e^{-1/2}+Ke)^2} \frac{\epsilon}{2\sqrt{d}} \left( \frac{dT}{6} - \frac{1}{3} \sum_{j=1}^{d} \max_{V \in \mathcal{V}_{d/4-1}} \left| \mathbb{E}_{V \cup \{j\}}\left[\widetilde{M}_j\right] - \mathbb{E}_V\left[\widetilde{M}_j\right] \right| \right)$$

$$\geq \frac{e^{-1/2}(K-1)}{(e^{-1/2}+Ke)^2} \frac{\epsilon}{2\sqrt{d}} \left( \frac{dT}{6} - \frac{1}{3} \sum_{j=1}^{d} T \cdot \sqrt{\frac{1}{2}\mathrm{KL}(\mathbb{P}_V \| \mathbb{P}_{V \cup \{j\}})} \right)$$

$$\geq \frac{e^{-1/2}(K-1)}{(e^{-1/2}+Ke)^2} \frac{\epsilon}{2\sqrt{d}} \left( \frac{dT}{6} - \frac{T\sqrt{d}}{3} \cdot \sqrt{\sum_{j=1}^{d} \frac{1}{2}\mathrm{KL}(\mathbb{P}_V \| \mathbb{P}_{V \cup \{j\}})} \right)$$

$$\geq \frac{e^{-1/2}(K-1)}{(e^{-1/2}+Ke)^2} \frac{\epsilon}{2\sqrt{d}} \left( \frac{dT}{6} - \frac{T\sqrt{d}}{3} \cdot \sqrt{\sum_{j=1}^{d} \frac{1}{2} C \cdot \frac{K}{(1+K)^2} \cdot \frac{\mathbb{E}_V\left[\widetilde{M}_j\right]\epsilon^2}{d}} \right)$$

$$\geq \frac{e^{-1/2}(K-1)}{(e^{-1/2}+Ke)^2} \frac{\epsilon}{2\sqrt{d}} \left( \frac{dT}{6} - \frac{T\sqrt{d}}{3} \cdot \sqrt{\frac{C}{8} \cdot \frac{K}{(1+K)^2} \cdot T\epsilon^2} \right) \qquad (\because \sum_{j=1}^{d} \mathbb{E}_V\left[\widetilde{M}_j\right] \leq \frac{dT}{4}).$$

By setting $\epsilon = \sqrt{\frac{d}{2CT} \cdot \frac{(1+K)^2}{K}}$, we finally have that

$$\sup_{\boldsymbol{\theta},\lambda} \mathbb{E}^{\pi}_{\boldsymbol{\theta},\lambda} \left[ \mathcal{R}(T|\boldsymbol{\theta},\lambda) \right] = \sup_{\boldsymbol{\theta},\lambda} \mathbb{E}^{\pi}_{\boldsymbol{\theta},\lambda} \left[ \sum_{t=1}^{T} R_t(S_t^*, \boldsymbol{\theta}, \lambda) - R_t(S_t, \boldsymbol{\theta}, \lambda) \right]$$

$$\geq \frac{e^{-1/2}(K-1)}{(e^{-1/2} + Ke)^2} \frac{\epsilon}{2\sqrt{d}} \left( \frac{dT}{6} - \frac{T\sqrt{d}}{3} \cdot \sqrt{\frac{C}{8} \cdot \frac{K}{(1+K)^2} \cdot T\epsilon^2} \right)$$

$$\geq \frac{e^{-1/2}(K-1)}{(e^{-1/2} + Ke)^2} \sqrt{\frac{1}{8CT} \cdot \frac{(1+K)^2}{K}} \left( \frac{dT}{6} - \frac{dT}{12} \right)$$

$$= \Omega\left( \frac{d\sqrt{T}}{\sqrt{K}} \right).$$

$\square$

## E.2 Technical Lemmas for Theorem 2

**Lemma 1.** *Fix $\epsilon \in (0, 1/d^{3/2})$, $\lambda = \frac{1}{2}$, and $V \in \mathcal{V}_{d/4}$, and define $\delta := d/4 - |\widetilde{U}_t \cap V|$. Then,*

$$\sum_{i \in S^*} p_t(i|S^*, \boldsymbol{\theta}_V, \lambda) - \sum_{i \in S^t} p_t(i|S_t, \boldsymbol{\theta}_V, \lambda) \geq \frac{e^{-1/2}(K-1)}{(e^{-1/2} + Ke)^2} \frac{\delta\epsilon}{2\sqrt{d}}.$$

*Proof of Lemma 1.* We split the proof into two cases: (i) $V \neq [d/4]$, and (ii) $V = [d/4]$.

**Case 1.** $V \neq [d/4]$.

We recall $x_{\widetilde{U}_t} := \operatorname{argmax}_{x_U \in S_t} x_U^\top \boldsymbol{\theta}_V$. If $\delta = 0$, i.e. $\widetilde{U}_t = V$, then the lemma holds trivially, thus we suppose that $\delta \neq 0$.

If $V \neq [d/4]$, it is obvious that $S^* = \{x_V, \dots, x_V\}$ with $g(S^*) = 1$, and so we have

$$\sum_{i \in S^*} p_t(i|S^*, \boldsymbol{\theta}_V, \lambda) - \sum_{i \in S^t} p_t(i|S_t, \boldsymbol{\theta}_V, \lambda)$$

$$\geq \frac{K \exp(x_V^\top \boldsymbol{\theta}_V)}{e^{-1/2} + K \exp(x_V^\top \boldsymbol{\theta}_V)} - \frac{K \exp(x_{\widetilde{U}_t}^\top \boldsymbol{\theta}_V)}{\exp^{-\lambda g(S_t)} + K \exp(x_{\widetilde{U}_t}^\top \boldsymbol{\theta}_V)}$$

$$\geq \frac{K \exp(x_V^\top \boldsymbol{\theta}_V)}{e^{-1/2} + K \exp(x_V^\top \boldsymbol{\theta}_V)} - \frac{K \exp(x_{\widetilde{U}_t}^\top \boldsymbol{\theta}_V)}{\exp^{-1/2} + K \exp(x_{\widetilde{U}_t}^\top \boldsymbol{\theta}_V)} \quad (\because g(S_t) \leq 1)$$

$$= \frac{e^{-1/2} K (\exp(x_V^\top \boldsymbol{\theta}_V) - \exp(x_{\widetilde{U}_t}^\top \boldsymbol{\theta}_V))}{(e^{-1/2} + K \exp(x_V^\top \boldsymbol{\theta}_V))(\exp^{-1/2} + K \exp(x_{\widetilde{U}_t}^\top \boldsymbol{\theta}_V))}$$

$$= \frac{e^{-1/2} K (\exp(x_V^\top \boldsymbol{\theta}_V) - \exp(x_{\widetilde{U}_t}^\top \boldsymbol{\theta}_V))}{(e^{-1/2} + Ke)^2} \quad (\because \exp(x_U^\top \boldsymbol{\theta}_V) \leq e, \forall U \in \mathcal{V}_{d/4})$$

$$\geq \frac{e^{-1/2} K ((x_V - X_{\widetilde{U}_t})^\top \boldsymbol{\theta}_V - (x_{\widetilde{U}_t}^\top \boldsymbol{\theta}_V)^2/2)}{(e^{-1/2} + Ke)^2} \quad (\because 1 + a \leq e^a \leq 1 + a + a^2/2, \forall a \in [0,1])$$

$$\geq \frac{e^{-1/2} K (\delta\epsilon/\sqrt{d} - (\sqrt{d}\epsilon)^2/2)}{(e^{-1/2} + Ke)^2}$$

$$\geq \frac{e^{-1/2} K \delta\epsilon}{2\sqrt{d}(e^{-1/2} + Ke)^2} \quad (\because (\sqrt{d}\epsilon)^2 \leq \epsilon/\sqrt{d} \leq \delta\epsilon/\sqrt{d})$$

$$> \frac{e^{-1/2}(K-1)\delta\epsilon}{2\sqrt{d}(e^{-1/2} + Ke)^2}.$$

**Case 2.** $V = [d/4]$.

We recall that $g(S) = 0$ if $S$ contains only $x_{U_0}$'s, and otherwise $g(S) = 1$. For $V = [d/4] = U_0$, since $g(\{x_0, \dots x_0\}) = 0$, we have to compare whether it is better to fill only $x_0$'s in $S^*$ or add another $x_{U'}$ in $S^*$, because $g(S^*)$ becomes 1 in the second case. Specifically, since the following inequality holds:

$$e^{\lambda g(S_0)} K \exp(x_0^\top \boldsymbol{\theta}_{U_0}) = K \exp(x_0^\top \boldsymbol{\theta}_{U_0})$$
$$< e^{1/2}(K-1) \exp(x_0^\top \boldsymbol{\theta}_{U_0})$$
$$< e^{\lambda g(\{x_{U_0}, \dots, x_{U_0}, x_{U'}\})} \left( (K-1) \exp(x_0^\top \boldsymbol{\theta}_{U_0}) + \exp(x_{U'}^\top \boldsymbol{\theta}_{U_0}) \right),$$

we obtain that $S^* = \{x_{U_0}, \dots, x_{U_0}, x_{U'}\}$ for $|U' \cap [d/4]| = d/4 - 1$, and $g(S^*) = 1$.

If $x_0 \in S_t$, then $\widetilde{U}_t = U_0 = V$. In this case $\delta = 0$, and the lemma holds trivially. Thus, we suppose that $x_0 \notin S_t$. Then, $g(S_t) = 1$, and we have that

$$\sum_{i \in S^*} p_t(i|S^*, \boldsymbol{\theta}_V, \lambda) - \sum_{i \in S^t} p_t(i|S_t, \boldsymbol{\theta}_V, \lambda)$$

$$\geq \left( 1 - \frac{e^{-1/2}}{e^{-1/2} + (K-1)\exp(x_0^\top \boldsymbol{\theta}_V) + \exp(x_{U'}^\top \boldsymbol{\theta}_V)} \right) - \left( 1 - \frac{e^{-1/2}}{e^{-1/2} + K \exp(x_{\widetilde{U}_t}^\top \boldsymbol{\theta}_V)} \right)$$

$$= e^{-1/2} \frac{(K-1)\exp(x_0^\top \boldsymbol{\theta}_V) + \exp(x_{U'}^\top \boldsymbol{\theta}_V) - K\exp(x_{\widetilde{U}_t}^\top \boldsymbol{\theta}_V))}{(e^{-1/2} + (K-1)\exp(x_0^\top \boldsymbol{\theta}_V) + \exp(x_{U'}^\top \boldsymbol{\theta}_V))(\exp^{-1/2} + K\exp(x_{\widetilde{U}_t}^\top \boldsymbol{\theta}_V))}$$

$$= e^{-1/2} \frac{(K-1)(\exp(x_0^\top \boldsymbol{\theta}_V) - \exp(x_{\widetilde{U}_t}^\top \boldsymbol{\theta}_V)) + (\exp(x_{U'}^\top \boldsymbol{\theta}_V) - \exp(x_{\widetilde{U}_t}^\top \boldsymbol{\theta}_V))}{(e^{-1/2} + Ke)^2}$$

Since $\widetilde{U}_t \neq V$ and $U'$ satisfies $|U' \cap V| = d/4 - 1$, we have that $\exp(x_{U'}^\top \boldsymbol{\theta}_V) - \exp(x_{\widetilde{U}_t}^\top \boldsymbol{\theta}_V) \geq 0$. Thus the right handside is bounded by

$$\frac{e^{-1/2}(K-1)(\exp(x_V^\top \boldsymbol{\theta}_V) - \exp(x_{\widetilde{U}_t}^\top \boldsymbol{\theta}_V))}{(e^{-1/2} + Ke)^2}$$

$$\geq \frac{v_0(K-1)((x_V - X_{\widetilde{U}_t})^\top \boldsymbol{\theta}_V - (x_{\widetilde{U}_t}^\top \boldsymbol{\theta}_V)^2/2)}{(e^{-1/2} + Ke)^2} \qquad (\because 1 + a \leq e^a \leq 1 + a + a^2/2, \forall a \in [0,1])$$

$$\geq \frac{v_0(K-1)(\delta\epsilon/\sqrt{d} - (\sqrt{d}\epsilon)^2/2)}{(e^{-1/2} + Ke)^2}$$

$$\geq \frac{v_0(K-1)\delta\epsilon}{2\sqrt{d}(e^{-1/2} + Ke)^2} \qquad (\because (\sqrt{d}\epsilon)^2 \leq \epsilon/\sqrt{d} \leq \delta\epsilon/\sqrt{d}).$$

$\square$

**Lemma 2** (Bound on KL divergence, Lemma D.2 of Lee and Oh(2024)). *For any $V \in \mathcal{V}_{d/4-1}$ and $j \in [d]$, there exists a positive constant $C > 0$ such that*

$$KL(\mathbb{P}_V \| \mathbb{P}_{V \cup \{j\}}) \leq C \cdot \frac{K}{(1+K)^2} \cdot \frac{\mathbb{E}_V \left[ \widetilde{M_j} \right] \epsilon^2}{d}$$

*Proof of Lemma 2.* In the proof of Lemma D.2 of Lee and Oh(2024), the following holds for some positive constant $C > 0$.

$$\mathrm{KL}(\mathbb{P}_V(\cdot \mid \widetilde{S}_t) \| \mathbb{P}_{V \cup \{j\}}(\cdot \mid \widetilde{S}_t)) \leq C \cdot \frac{v_0 K}{(v_0 + K)^2} \cdot \frac{m_j(\widetilde{S}_t)\epsilon^2}{d},$$

where $m_j(\widetilde{S}_t) := \mathbb{1}\{j \in \widetilde{U}_t\}$, and $v_0$ is a outside option parameter, defined as $\exp(-\lambda g(\widetilde{S}_t))$ in our setting. Since $g(\widetilde{S}_t)$ has a value of 0 or 1, we have that

$$\mathrm{KL}(\mathbb{P}_V(\cdot \mid \widetilde{S}_t) \| \mathbb{P}_{V \cup \{j\}}(\cdot \mid \widetilde{S}_t)) \leq C \cdot \frac{K}{(1+K)^2} \cdot \frac{m_j(\widetilde{S}_t)\epsilon^2}{d},$$

Therefore, by the chain rule of relative entropy, we have that

$$
\begin{aligned}
\mathrm{KL}(\mathbb{P}_V || \mathbb{P}_{V \cup \{j\}}) &= \sum_{t=1}^{T} \mathbb{E}_V \left[ \mathrm{KL}(\mathbb{P}_V(\cdot \mid \widetilde{S}_t) \, || \, \mathbb{P}_{V \cup \{j\}}(\cdot \mid \widetilde{S}_t)) \right] \\
&\leq \sum_{t=1}^{T} C \cdot \frac{K}{(1+K)^2} \cdot \frac{\mathbb{E}_V \left[ m_j(\widetilde{S}_t) \right] \epsilon^2}{d} \\
&= C \cdot \frac{K}{(1+K)^2} \cdot \frac{\mathbb{E}_V \left[ \widetilde{M}_j \right] \epsilon^2}{d}.
\end{aligned}
$$

$\square$

# F  REGRET UPPER BOUND OF ALGORITHM 1 (`OFU-DMNL`)

## F.1  TECHNICAL LEMMAS

In this section, we introduce technical lemmas used to derive the regret bound of Algorithm 1.

**Lemma 3.** *Suppose Assumptions 1 and 3 hold, and set $\nu = \frac{\omega}{2}$ in Algorithm 1. Let us define the event $\mathcal{T}^e$ as the set of rounds corresponding to adaptive exploration, as follows:*

$$
\mathcal{T}^e := \left\{ t \in [T] : \|[\mathbf{0_d}, 1]\|_{\mathbf{H}_t^{-1}} > \frac{\omega \hat{\lambda}_t}{2\alpha_t} \right\}. \tag{14}
$$

*Then, for $t \notin \mathcal{T}^e$, $\widetilde{f}_t$ is monotone and submodular where $\widetilde{f}_t$ is defined as follows:*

$$
\widetilde{f}_t(S) := \log \left( \sum_{i \in S} \exp \left( \mathbf{z}_{ti}(S)^\top \mathbf{w}_t + \alpha_t \|\mathbf{z}_{ti}(S)\|_{\mathbf{H}_t^{-1}} \right) \right) = \log \left( \sum_{i \in S} \exp \left( \mathrm{ucb}(\mathbf{z}_{ti}(S)) \right) \right).
$$

*Proof of Lemma 3.* Suppose $\|[\mathbf{0_d}, 1]\|_{\mathbf{H}_t^{-1}} \leq \frac{\omega \hat{\lambda}_t}{2\alpha_t}$ holds.

**Monotonicity.** Recall that the diversity-augmented feature vector is defined $\mathbf{z}_{ti}(S) := [\mathbf{x}_{ti}, g_t(S)]$. Then, $\widetilde{f}_t(S \cup \{i\}) - \widetilde{f}_t(S)$ can be written as follows:

$$
\widetilde{f}_t(S \cup \{i\}) - \widetilde{f}_t(S) = \log \left[ \frac{\sum_{j \in S} \exp \left( \mathrm{ucb}([\mathbf{x}_{tj}, g_t(S \cup \{i\})]) \right) + \exp \left( \mathrm{ucb}([\mathbf{x}_{ti}, g_t(S \cup \{i\})]) \right)}{\sum_{j \in S} \exp \left( \mathrm{ucb}([\mathbf{x}_{tj}, g_t(S)]) \right)} \right].
$$

Then, for each $j \in S$ we have

$$
\begin{aligned}
&\mathrm{ucb}([\mathbf{x}_{tj}, g_t(S \cup \{i\})]) - \mathrm{ucb}(\mathbf{x}_{tj}, g_t(S)) \\
&= \hat{\lambda}_t(g_t(S \cup \{i\}) - g_t(S)) + \alpha_t \left( \|[\mathbf{x}_{tj}, g_t(S \cup \{i\})]\|_{\mathbf{H}_t^{-1}} - \|[\mathbf{x}_{tj}, g_t(S)]\|_{\mathbf{H}_t^{-1}} \right) \\
&\geq \hat{\lambda}_t(g_t(S \cup \{i\}) - g_t(S)) - \alpha_t \|[\mathbf{0}_d, g_t(S \cup \{i\}) - g_t(S)]\|_{\mathbf{H}_t^{-1}} \\
&= \left( \hat{\lambda}_t - \alpha_t \|[\mathbf{0}_d, 1]\|_{\mathbf{H}_t^{-1}} \right) (g_t(S \cup \{i\}) - g_t(S)) \\
&\geq \left( 1 - \frac{\omega}{2} \right) \hat{\lambda}_t(g_t(S \cup \{i\}) - g_t(S)) \geq 0,
\end{aligned}
$$

where the last inequality holds since $\|[\mathbf{0_d}, 1]\|_{\mathbf{H}_t^{-1}} \leq \frac{w \hat{\lambda}_t}{2\alpha_t}$. Therefore, we conclude $\widetilde{f}_t(S \cup \{i\}) - \widetilde{f}_t(S) > 0$.

**Submodularity.** To show submodularity of $\widetilde{f}_t$, it is enough to show that the inequality in Eq. 9 holds. If $g_t(S) = g_t(S_2)$, then $g_t(S_1) - g_t(S) \geq g_t(S_3) - g_t(S_2)$ by submodularity of $g_t$, and so $g_t(S_1) \leq g_t(S_3)$. By the monotonicity of $g_t$, we have $g_t(S_1) = g_t(S_3)$. Therefore, the inequality in Eq. 9 holds.

Now, suppose $g_t(S) < g_t(S_2)$. Then, for all $j_1 \in S$ and $j_2 \in S_3$, we have

$$\hat{\lambda}_t \left( g_t(S_1) + g_t(S_2) - g_t(S) - g_t(S_3) \right)$$
$$+ \alpha_t \left( \|[\mathbf{x}_{t,j_1}, g_t(S_1)]\|_{\mathbf{H}_t^{-1}} + \|[\mathbf{x}_{t,j_2}, g_t(S_2)]\|_{\mathbf{H}_t^{-1}} - \|[\mathbf{x}_{t,j_1}, g_t(S))\|_{\mathbf{H}_t^{-1}} - \|[\mathbf{x}_{t,j_2}, g_t(S))\|_{\mathbf{H}_t^{-1}} \right)$$
$$\geq \hat{\lambda}_t \left( g_t(S_1) - g_t(S) - (g_t(S_3) - g_t(S_2)) \right) - \alpha_t(g_t(S_1) - g_t(S)) \cdot \|[\mathbf{0}_d, 1]\|_{\mathbf{H}_t^{-1}}$$
$$\quad - \alpha_t(g_t(S_3) - g_t(S_2)) \cdot \|[\mathbf{0}_d, 1]\|_{\mathbf{H}_t^{-1}}$$
$$\geq \hat{\lambda}_t \omega (g_t(S_1) - g_t(S)) - 2\alpha_t(g_t(S_1) - g_t(S)) \cdot \|[\mathbf{0}_d, 1]\|_{\mathbf{H}_t^{-1}}$$
$$> \left( \hat{\lambda}_t \omega - 2\alpha_t \frac{\hat{\lambda}_t \omega}{2\alpha_t} \right) (g_t(S_1) - g_t(S)) \geq 0 \,.$$

$\square$

**Lemma 4.** *Suppose that Assumptions 1 and 3 hold. If $\lambda_{\min}(\mathbf{H}_t) \geq \frac{\alpha_T}{l}\left(1 + \frac{2}{\omega}\right)$, we have $t \notin \mathcal{T}^e$.*

*Proof of Lemma 4.* Suppose that $\lambda_{\min}(\mathbf{H}_t) \geq \frac{\alpha_T(\delta)}{l}\left(1 + \frac{2}{\omega}\right)$. To show $t \notin \mathcal{T}^e$, we have to prove that $\|[\mathbf{0}, 1]\|_{\mathbf{H}_t^{-1}} \leq \frac{\omega \hat{\lambda}_t}{2\alpha_t(\delta)}$. By the properties of eigenvalues,

$$\|[\mathbf{0_d}, 1]\|_{\mathbf{H}_t^{-1}} \leq \lambda_{\max}(\mathbf{H}_t^{-1}) = \frac{1}{\lambda_{\min}(\mathbf{H}_t)} \leq l \frac{\omega}{(2+\omega)\alpha_T(\delta)} \leq \lambda^* \frac{\omega}{(2+\omega)\alpha_t(\delta)} \,.$$

Since $\lambda^* = [\mathbf{0_d}, 1]^\top [\boldsymbol{\theta}^*, \lambda^*] \leq [\mathbf{0_d}, 1]^\top (\hat{\boldsymbol{\theta}}_t, \hat{\lambda}_t) + \alpha_t \|[\mathbf{0_d}, 1]\|_{\mathbf{H}_t^{-1}} \leq \hat{\lambda}_t + \alpha_t \|[\mathbf{0_d}, 1]\|_{\mathbf{H}_t^{-1}}$ by Lemma 7, then we have

$$\|[\mathbf{0_d}, 1]\|_{\mathbf{H}_t^{-1}} \leq \left( \hat{\lambda}_t + \alpha_t \|(\mathbf{0}, 1)\|_{\mathbf{H}_t^{-1}} \right) \cdot \frac{\omega}{(2+\omega)\alpha_t}$$
$$= \frac{\omega}{(2+\omega)\alpha_t} \hat{\lambda}_t + \frac{\omega}{2+\omega} \|[\mathbf{0_d}, 1]\|_{\mathbf{H}_t^{-1}} \,.$$

By subtracting $\frac{\omega}{2+\omega} \|[\mathbf{0_d}, 1]\|_{\mathbf{H}_t^{-1}}$ on the both side, it follows that

$$\left( 1 - \frac{\omega}{2+\omega} \right) \|[\mathbf{0_d}, 1]\|_{\mathbf{H}_t^{-1}} \leq \frac{\omega}{(2+\omega)\alpha_t(\delta)} \lambda_t,$$

and consequently,

$$\|[\mathbf{0_d}, 1]\|_{\mathbf{H}_t^{-1}} \leq \frac{\omega \lambda_t}{2\alpha_t} \,.$$

$\square$

**Lemma 5.** *Suppose Assumptions 1 and 2 hold. Let $\tau := |\mathcal{T}^e \cap [t]|$ denote the number of adaptive exploration rounds up to round $t$. Then, with probability at least $1 - (d+1)\exp\left(-\frac{\tau \sigma_0}{10}\right)$, we have:*

$$\lambda_{\min} \left( \sum_{t'=1}^{t} \sum_{i \in S_{t'}} \mathbf{z}_{t'i}(S_{t'}) \mathbf{z}_{t'i}(S_{t'})^\top \right) \geq \frac{\tau K \sigma_0}{2} \,,$$

*where $\sigma_0$ is defined in Eq.(7).*

*Proof of Lemma 5.* Let $\mathcal{H}_t$ be the history $\{\{X_{t'}\}_{t' \in [t]}, \{S_{t'}\}_{t' \in [t]}, \{y_{t'}\}_{t' \in [t]}\}$ until round $t$. By Assumption 2, for any adaptive exploration round $t' \in \mathcal{T}^e \cap [t]$, we have

$$
\lambda_{\min}\left(\mathbb{E}\left[\sum_{i \in S_{t'}} \mathbf{z}_{t'i}(S_{t'})\mathbf{z}_{t'i}(S_{t'})^\top | \mathcal{H}_{t'-1}\right]\right) = \lambda_{\min}\left(\mathbb{E}\left[\sum_{i \in S_{t'}} \mathbf{z}_{t'i}(S_{t'})\mathbf{z}_{t'i}(S_{t'})^\top\right]\right)
$$
$$
= \lambda_{\min}\left(\frac{1}{|\mathcal{S}|}\sum_{S \in \mathcal{S}}\left[\sum_{i \in S}\mathbf{z}_{t'i}(S_{t'})\mathbf{z}_{t'i}(S_{t'})^\top\right]\right)
$$
$$
\geq K\sigma_0.
$$

Then, by the subadditivity of minimum eigenvalues,

$$
\lambda_{\min}\left(\sum_{t'=1}^{t}\mathbb{E}\left[\sum_{i \in S_{t'}}\mathbf{z}_{t'i}(S_{t'})\mathbf{z}_{t'i}(S_{t'})^\top | \mathcal{H}_{t'-1}\right]\right) \geq \lambda_{\min}\left(\sum_{t' \in \mathcal{T}^e \cap [t]}\mathbb{E}\left[\sum_{i \in S_{t'}}\mathbf{z}_{t'i}(S_{t'})\mathbf{z}_{t'i}(S_{t'})^\top | \mathcal{H}_{t'-1}\right]\right)
$$
$$
\geq \sum_{t' \in \mathcal{T}^e \cap [t]}^{t}\lambda_{\min}\left(\mathbb{E}\left[\sum_{i \in S_{t'}}\mathbf{z}_{t'i}(S_{t'})\mathbf{z}_{t'i}(S_{t'})^\top | \mathcal{H}_{t'-1}\right]\right)
$$
$$
\geq |\mathcal{T}^e \cap [t]| \cdot K\sigma_0
$$
$$
= \tau K\sigma_0
$$

In other words, $\mathbb{P}\left[\lambda_{\min}\left(\sum_{t'=1}^{t}\mathbb{E}\left[\sum_{i \in S_{t'}}\mathbf{z}_{t'i}(S_{t'})\mathbf{z}_{t'i}(S_{t'})^\top | \mathcal{H}_{t'-1}\right]\right) \geq \tau K\sigma_0\right)\right] = 1$ holds.

By applying Lemma 8 and $\lambda_{\max}\left(\sum_{i \in S_{t'}}\mathbf{z}_{t'i}(S_{t'})\mathbf{z}_{t'i}(S_{t'})^\top\right) \leq K$ for all $t' \in [t]$ to compute the lower bound of the minimum eigenvalue of the Gram matrix after $t$ rounds, we have

$$
\mathbb{P}\left[\lambda_{\min}\left(\sum_{t'=1}^{t}\sum_{i \in S_{t'}}\mathbf{z}_{t'i}(S_{t'})\mathbf{z}_{t'i}(S_{t'})^\top\right) \leq \frac{\tau K\sigma_0}{2}\right] \leq (d+1)\left(\frac{e^{0.5}}{0.5^{0.5}}\right)^{-\frac{\tau K\sigma_0}{K}} \leq (d+1)e^{-\frac{\tau\sigma_0}{10}},
$$

using the fact that $-0.5 - 0.5\log(0.5) \leq -0.1$. $\qquad\square$

**Lemma 6.** *Suppose that Assumptions 1, 2, and 3 hold. If we set $\nu = \frac{\omega}{2}$ in Algorithm 1, then for $\delta \in (0,1)$ with probability $1 - \delta - (d+1)T^{-\mathcal{O}(\frac{\sigma_0\sqrt{d}\log K}{\kappa K l_\lambda \omega})}$, the total number of adaptive exploration rounds is bounded as follows:*

$$
|\mathcal{T}^e| = \mathcal{O}\left(\frac{\sqrt{d}\log T\log K}{\kappa K\sigma_0\omega l}\right).
$$

*Proof of Lemma 6.* We will show that the number of adaptive exploration rounds can not exceed $\frac{2\alpha_T}{\kappa l K\sigma_0}\left(1 + \frac{2}{\omega}\right)$ rounds by contradiction. Suppose Algorithm 1 induces $\frac{2\alpha_T}{\kappa l K\sigma_0}\left(1 + \frac{2}{\omega}\right)$ adaptive exploration rounds. Then, by Lemma 5, with probability at least $1 - (d+1)\exp\left(-\frac{2\alpha_T\left(1+\frac{2}{\omega}\right)}{10\kappa l K}\right) = 1 - (d+1)T^{-\mathcal{O}(\frac{\sigma_0\sqrt{d}\log K}{\kappa K\omega l})}$, we have

$$
\lambda_{\min}(V_{t+1}) \geq \frac{1}{\kappa}\frac{\alpha_T}{l}\left(1 + \frac{2}{\omega}\right),
$$

where $\mathbf{V}_t := \sum_{t'=1}^{t-1}\sum_{i \in S_{t'}}\mathbf{z}_{t'i}(S_{t'})\mathbf{z}_{t'i}(S_{t'})^\top$. Note that for any $\mathbf{x}_i, \mathbf{x}_j \in \mathbb{R}^d$, $(\mathbf{x}_i - \mathbf{x}_j)(\mathbf{x}_i - \mathbf{x}_j)^\top = \mathbf{x}_i\mathbf{x}_i^\top + \mathbf{x}_j\mathbf{x}_j^\top - \mathbf{x}_i\mathbf{x}_j^\top - \mathbf{x}_j\mathbf{x}_i^\top \succeq \mathbf{0}_{d \times d}$, which implies $\mathbf{x}_i\mathbf{x}_i^\top + \mathbf{x}_j\mathbf{x}_j^\top \succeq \mathbf{x}_i\mathbf{x}_j^\top + \mathbf{x}_j\mathbf{x}_i^\top$. To simplify, for $i \in S_t$, if we abbreviate $p_t(i \mid S_t, \mathbf{w}_t)$ by $p_{ti}(\mathbf{w}_t)$ and $\mathbf{z}_{ti}(S_t)$ by $\mathbf{z}_{ti}$, then for all

$s \in [t-1]$, we have

$$
\begin{aligned}
\mathcal{G}_s(\mathbf{w}_{s+1}) &= \sum_{i \in S_s} p_{si}(\mathbf{w}_{s+1}) \mathbf{z}_{si} \mathbf{z}_{si}^\top - \sum_{i \in S_s} \sum_{j \in S_s} p_{si}(\mathbf{w}_{s+1}) p_{sj}(\mathbf{w}_{s+1}) \mathbf{z}_{si} \mathbf{z}_{sj}^\top \\
&= \sum_{i \in S_s} p_{si}(\mathbf{w}_{s+1}) \mathbf{z}_{si} \mathbf{z}_{si}^\top - \tfrac{1}{2} \sum_{i \in S_s} \sum_{j \in S_s} p_{si}(\mathbf{w}_{s+1}) p_{sj}(\mathbf{w}_{s+1}) \big( \mathbf{z}_{si} \mathbf{z}_{sj}^\top + \mathbf{z}_{sj} \mathbf{z}_{si}^\top \big) \\
&\succeq \sum_{i \in S_s} p_{si}(\mathbf{w}_{s+1}) \mathbf{z}_{si} \mathbf{z}_{si}^\top - \tfrac{1}{2} \sum_{i \in S_s} \sum_{j \in S_s} p_{si}(\mathbf{w}_{s+1}) p_{sj}(\mathbf{w}_{s+1}) \big( \mathbf{z}_{si} \mathbf{z}_{si}^\top + \mathbf{z}_{sj} \mathbf{z}_{sj}^\top \big) \\
&= \sum_{i \in S_s} p_{si}(\mathbf{w}_{s+1}) \mathbf{z}_{si} \mathbf{z}_{si}^\top - \sum_{i \in S_s} \sum_{j \in S_s} p_{si}(\mathbf{w}_{s+1}) p_{sj}(\mathbf{w}_{s+1}) \big( \mathbf{z}_{si} \mathbf{z}_{si}^\top \big) \\
&= \sum_{i \in S_s} p_{si}(\mathbf{w}_{s+1}) \left( 1 - \sum_{j \in S_s} p_{sj}(\mathbf{w}_{s+1}) \right) \mathbf{z}_{si} \mathbf{z}_{si}^\top \\
&= \sum_{i \in S_s} p_{si}(\mathbf{w}_{s+1}) p_{s0}(\mathbf{w}_{s+1}) \mathbf{z}_{si} \mathbf{z}_{si}^\top \\
&\succeq \kappa \sum_{i \in S_s} \mathbf{z}_{si} \mathbf{z}_{si}^\top ,
\end{aligned}
$$

where $\kappa := \min_{t \in [T], S \in \mathcal{S}, i \in S, \|\boldsymbol{\theta}\|_2 \leq 1, 0 \leq \lambda \leq 1} p_t(i|S, \boldsymbol{\theta}, \lambda) p_t(0|S, \boldsymbol{\theta}, \lambda) > 0$. Hence, we have

$$
\mathbf{H}_{t+1} = \Lambda \mathbf{I}_{d+1} + \sum_{s=1}^{t-1} \mathcal{G}_s(\mathbf{w}_{s+1}) \succeq \Lambda \mathbf{I}_{d+1} + \kappa \sum_{s=1}^{t-1} \sum_{i \in S_s} \mathbf{z}_{si} \mathbf{z}_{si}^\top \succeq \kappa \mathbf{V}_{t+1} .
$$

Thus, it holds that

$$
\lambda_{\min}(\mathbf{H}_{t+1}) \geq \kappa \lambda_{\min}(\mathbf{V}_{t+1}) \geq \frac{\alpha}{l} \left( 1 + \frac{2}{\omega} \right) .
$$

By Lemma 4, this implies $t + 1 \notin \mathcal{T}^e$. Therefore, with probability at least $1 - \delta - (d + 1) T^{-\mathcal{O}(\frac{\sigma_0 \sqrt{d} \log K}{\kappa K \omega l})}$, the number of adaptive exploration rounds is bounded by

$$
|\mathcal{T}^e| \leq \frac{2\alpha_T}{\kappa l K \sigma_0} \left( 1 + \frac{2}{\omega} \right) = \mathcal{O} \left( \frac{\sqrt{d} \log T \log K}{\kappa K \sigma_0 \omega l} \right) .
$$

$\square$

### F.2 PROOF OF THEOREM 3

*Proof of Theorem 3.* We define the event $\mathcal{T}^e$ as the set of rounds corresponding to adaptive exploration, formally defined in Equation 14.

For $t \notin \mathcal{T}^e$, by Lemma 3 since $\widetilde{f}_t$ in Eq.(8) is monotone and submodular, we have

$$
\begin{aligned}
R(S_t^*, \boldsymbol{\theta}^*, \lambda^*) &= \frac{\exp(f_t(S_t^*))}{1 + \exp(f_t(S_t^*))} \\
&\leq \frac{\exp(\widetilde{f}_t(S_t^*))}{1 + \exp(\widetilde{f}_t(S_t^*))} \\
&\leq \frac{\exp\left( (\frac{e}{e-1}) \widetilde{f}_t(S_t)) \right)}{1 + \exp\left( (\frac{e}{e-1}) \widetilde{f}_t(S_t)) \right)\}} \qquad (\because \widetilde{f}_t \text{ is submodular}) \\
&= \frac{\left[ \exp \widetilde{f}_t(S_t)) \right]^{(\frac{e}{e-1})}}{1 + \left[ \exp \widetilde{f}_t(S_t)) \right]^{(\frac{e}{e-1})}} \\
&\leq \left( \frac{e+1}{e} \right) \widetilde{R}_t(S_t) ,
\end{aligned}
$$

where the last inequality holds because $h(\psi) := \left(\frac{\psi^\alpha}{1+\psi^\alpha}\right) / \left(\frac{\psi}{1+\psi}\right) < 1 + \frac{1}{e}$ holds for $\psi = \exp(\widetilde{f}_t(S_t))$ and $\alpha = \frac{e}{e-1}$ (refer to the proof of Theorem 1).

Therefore, with probability $1 - \delta - (d+1)T^{-\mathcal{O}(\frac{\sigma_0\sqrt{d}\log K}{\kappa K l \omega})}$,

$$
\begin{aligned}
\mathcal{R}^\gamma(T) &= \sum_{t=1}^{T} \mathbb{E}[\gamma R_t(S_t^*, \boldsymbol{\theta}^*, \lambda^*) - R_t(S_t, \boldsymbol{\theta}^*, \lambda^*)] \\
&\leq \sum_{t \notin \mathcal{T}^e} \mathbb{E}[\gamma R_t(S_t^*, \boldsymbol{\theta}^*, \lambda^*) - R_t(S_t, \boldsymbol{\theta}^*, \lambda^*) + |\mathcal{T}^e|] \\
&\leq \sum_{t \notin \mathcal{T}^e} \mathbb{E}[\widetilde{R}_t(S_t) - R_t(S_t, \boldsymbol{\theta}^*, \lambda^*)] + |\mathcal{T}^e| \\
&= \widetilde{\mathcal{O}}\left(\frac{\sqrt{K}(d+1)}{K+1} \cdot \sqrt{T} + \frac{1}{\kappa}(d+1)^2\right) + \mathcal{O}\left(\frac{\sqrt{d}\log T \log K}{\kappa K \sigma_0 \omega l}\right) \quad (15) \\
&= \widetilde{\mathcal{O}}\left(\frac{\sqrt{K}(d+1)}{K+1} \cdot \sqrt{T} + \frac{1}{\kappa}(d+1)^2 + \frac{\sqrt{d}}{\kappa K \sigma_0 \omega l}\right),
\end{aligned}
$$

where for Eq.(15), we invoke the regret bound of `OFU-MNL+` under uniform revenue setting from Lee & Oh (2024), as our diversified optimistic expected reward $\widetilde{R}(S_t)$ serves as an optimistic estimate of $R_t(S_t, \boldsymbol{\theta}^*, \lambda^*)$. The subsequent analysis closely follows the proof of Theorem 2 in Lee & Oh (2024). □

## G  EXPERIMENTAL DETAILS

### G.1  BASELINES

We compare the empirical performance of our algorithm against the existing MNL bandit algorithms `UCB-MNL`, `TS-MNL`, and `OFU-MNL+`, along with two variants of `OFU-MNL+` that incorporate assortment diversity, in the DMNL bandit setting.

**Algorithm for MNL bandits with submodular rewards.**  We consider the item-wise optimistic construction algorithm `OFU-MNL-DR` (Algorithm 2) for the original MNL choice model with a submodular reward function as a baseline. Inspired by Qin et al. (2014), diversity in the original MNL bandit can be promoted by modifying only the reward function as $R'_t(S, \boldsymbol{\theta}^*, \lambda) := \frac{\sum_{j \in S} \exp(\mathbf{x}_{tj}^\top \boldsymbol{\theta}^*)}{1 + \sum_{j \in S} \exp(\mathbf{x}_{tj}^\top \boldsymbol{\theta}^*)} + \lambda g(S)$, where $g(S)$ is the diversity score function, and $\lambda$ is a balancing parameter between relevance and diversity. We adapt `OFU-MNL+` with greedy assortment construction to maximize $R'_t$. Notably, `OFU-MNL+` requires the value of $\lambda$ to be specified as a hyperparameter.

---

**Algorithm 2** `OFU-MNL-DR` (`OFU-MNL-D`iversity integrated **R**eward)

1: **Input:**  diversity function $\{g_t\}_{t\geq1}$, regularization parameter $\Lambda$, confidence radius $\{\alpha_t\}_{t\geq1}$, step size $\eta$, balancing diversity parameter $\lambda$
2: **Initialization:** $\mathbf{H}_1 = \Lambda \mathbf{I}_d$ and $\boldsymbol{\theta}_1$ at any point in $\{\boldsymbol{\theta} \in \mathbb{R}^d : \|\boldsymbol{\theta}\|_2 \leq 1\}$
3: **for** $t = 1, \ldots, T$ **do**
4:     Compute $u_{t,i} = \mathbf{x}_{t,i}^\top \boldsymbol{\theta}_t + \alpha_t \|\mathbf{x}_{t,i}\|_{\mathbf{H}_t^{-1}}$
5:     $S_t \leftarrow \emptyset$
6:     **for** $k = 1, \ldots, K$ **do**
7:         $a_{t,k} \leftarrow \text{argmax}_{e \in [N] \setminus S_t} \left[ \frac{\sum_{i \in S_t \cup \{e\}} \exp(u_{t,i})}{1 + \sum_{i \in S_t \cup \{e\}} \exp(u_{t,i})} + \lambda g(S_t \cup \{e\}) \right]$
8:         $S_t \leftarrow S_t \cup \{a_{t,k}\}$
9:     Offer $S_t$ and observe $y_t$
10:    Update $\widetilde{\mathbf{H}}_t = \mathbf{H}_t + \eta\, \mathcal{G}_t(\boldsymbol{\theta}_{t+1})$, and update the estimator $\boldsymbol{\theta}_{t+1}$
11:    Update $\mathbf{H}_{t+1} = \mathbf{H}_t + \mathcal{G}_t(\mathbf{w}_{t+1})$

---

**Exhaustive-Search Algorithm for DMNL bandits.** We also consider the exhaustive-search algorithm, referred to as `OFU-DMNL-FULL` (Algorithm 3), for the DMNL bandit model. This algorithm evaluates all $\binom{N}{K}$ possible assortments in each round and thus requires approximately $\mathcal{O}\left(\left(\frac{eN}{K}\right)^K\right)$ reward estimations per round.

---

**Algorithm 3** `OFU-DMNL-FULL` (`OFU-DMNL` with exhaustive-search)

---

1: **Input:** diversity function $\{g_t\}_{t\geq 1}$, regularization parameter $\Lambda$, confidence radius $\{\alpha_t\}_{t\geq 1}$, step size $\eta$, exploration parameter $\nu$
2: **Initialization:** $\mathbf{H}_1 = \Lambda\mathbf{I}_{d+1}$ and $\mathbf{w}_1$ at any point in $\mathcal{W}$.
3: **for** $t = 1, \ldots, T$ **do**
4:    Offer $S_t \leftarrow \underset{S\in\mathcal{S}}{\operatorname{argmax}} \widetilde{R}_t(S)$, and observe $y_t$
5:    Update $\widetilde{\mathbf{H}}_t = \mathbf{H}_t + \eta\,\mathcal{G}_t(\mathbf{w}_t)$, $\mathbf{w}_{t+1}$, and $\mathbf{H}_{t+1} = \mathbf{H}_t + \mathcal{G}_t(\mathbf{w}_{t+1})$

---

## G.2 EXPERIMENTAL RESULTS IN DIVERSE ENVIRONMENTS

### G.2.1 RUNTIME COMPARISON

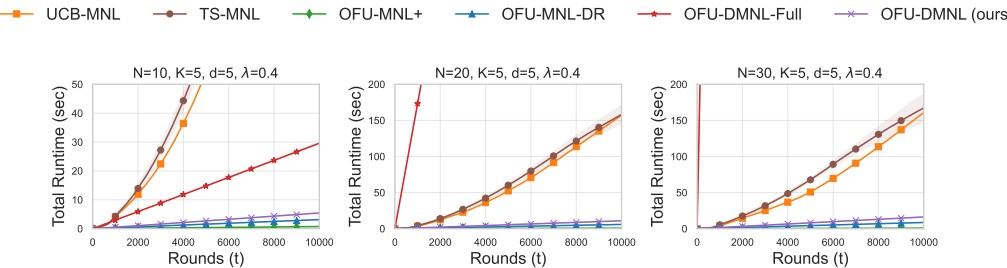

Figure 3: Cumulative runtime of algorithms under $N = 10, 20, 30$ and $T = 1000$

The results in Figure 3 shows that our algorithm operates very efficiently by leveraging online mirror descent. In particular, compared to the exhaustive-search algorithm, which requires per-round $O((eN/K)^K)$ computation, our algorithm runs in $O(NK)$ time and thus achieves incomparably better performance in large-$N$ settings. Although it is slower than other OMD-based MNL algorithms due to the cost of item-wise construction, it is still faster than MLE-based methods such as `UCB-MNL` and `TS-MNL`. This demonstrates that, despite incorporating the estimation of the diversity parameter, our algorithm remains computationally efficient.

Among the baselines, `OFU-MNL+` outperforms `UCB-MNL` and `TS-MNL`, while `OFU-DMNL-FULL` is entirely impractical from a runtime perspective. Therefore, in the subsequent experiments we focus on comparing the performance of the three algorithms `OFU-MNL+`, `OFU-MNL-DR`, and `OFU-DMNL`.

### G.2.2 REGRET COMPARISON

As shown in Figure 4, our algorithm demonstrates strong performance even when $N$ and $K$ are large. Furthermore, Figure 5 shows that as the balance between $\|\boldsymbol{\theta}^*\|_2$ and $\lambda^*$ varies across $0.6 : 0.4$, $0.5 : 0.5$, $0.4 : 0.6$, and $0.3 : 0.7$, the relative ranking of `OFU-MNL` and `OFU-MNL-DR` fluctuates, whereas our proposed algorithm consistently adapts and learns robustly across all balance settings.

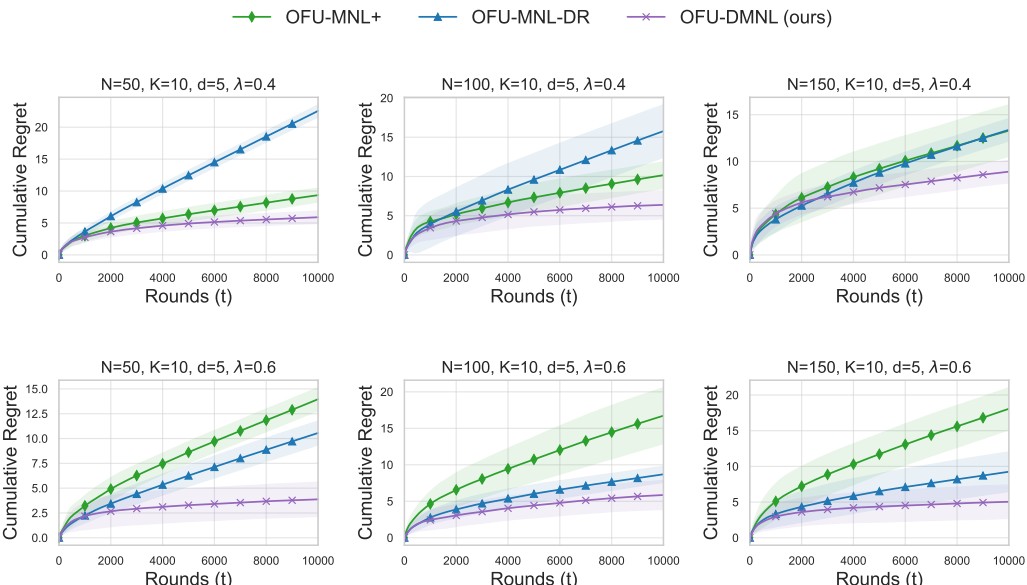

Figure 4: Performance of algorithms under various $(N, K, \lambda^*)$ configurations

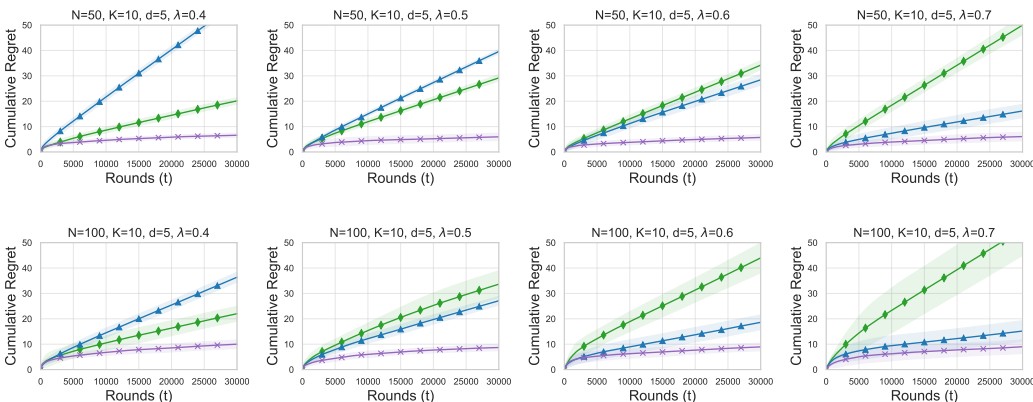

Figure 5: Performance of algorithms under different balances between relevance and diversity

### G.3  EXPERIMENT BASED ON REAL-WORLD DATA

We additionally designed a semi-synthetic experiment based on a real-world dataset, which provides the most practical and feasible alternative in the absence of access to online field deployment. We used the Massive Rotten Tomatoes Movie & Review dataset provided in Kaggle (https://www.kaggle.com), which contains over $1.4$M+ reviews on $140$K+ unique movies, each labeled as positive or negative, along with rich movie-level metadata. We first converted each review text into a vector representation using TF-IDF (via TfidfVectorizer from scikit-learn), followed by dimensionality reduction using truncated SVD to obtain $d$-dimensional context vectors. This process resulted in a context–label dataset suitable for downstream modeling. From this dataset, we trained a linear model to classify the binary labels, which was then used to approximate the true relevance utility of each movie from its context features. This constructed an online assortment selection environment in which we evaluated the performance of standard MNL bandit baselines, their variants, and our proposed method.

In each round of the online experiment, we $N$ randomly sampled movies , and asked the algorithm to choose an assortment of size $K$. We use exponential decaying categorical function defined as $g(S) := 1 + \rho + \ldots + \rho^{n_S - 1}$, where $n_S$ is the number of categories covered by the assortment $S$.

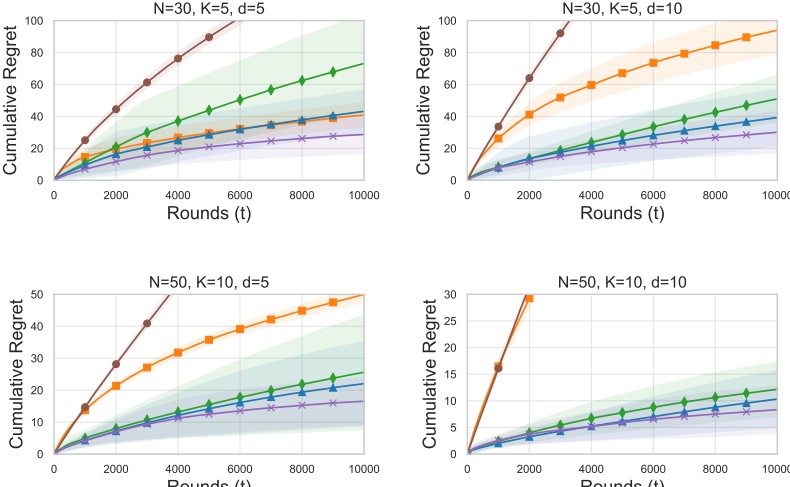

Figure 6: Performance of algorithms with synthetic data

As shown in Figure 6, our algorithm performs robustly in the synthetic-data experiments and consistently outperforms the baseline algorithms. These results suggest that our method is also likely to perform well on real-world data.

## H  AUXILIARY LEMMAS

**Proposition 2** (Proposition B.6 in Bach (2013)). *Let $f$ be a monotone submodular function and $\phi$ a non-decreasing concave function. Then, the composition function $\phi(f(S))$ is submodular.*

*Proof of Proposition 2.* Define $h(S) := \phi(f(S))$. By submodularity of $f$, for any $S_1 \subseteq S_2$ and $e \notin S_2$, we have

$$f(S_1 \cup \{e\}) - f(S_1) \geq f(S_2 \cup \{e\}) - f(S_2).$$

Also, concavity and non-decreasing property of $\phi$ imply that the difference $\psi(x, \delta) = \phi(x + \delta) - \phi(x)$ is non-increasing in the $x$ and non-decreasing in $\delta$. That is, if $x_1 \leq x_2$, then $\psi(x_1, \delta) \geq$

$\psi(x_2, \delta)$, and if $\delta_1 \le \delta_2$, then $\psi(x, \delta_1) \le \psi(x, \delta_2)$. Using this, we obtain the following:

$$
\begin{aligned}
\phi(f(S_1 \cup \{e\})) - \phi(f(S_1)) &= \phi(f(S_1) + f(S_1 \cup \{e\}) - f(S_1)) - \phi(f(S_1)) \\
&= \psi(f(S_1), f(S_1 \cup \{e\}) - f(S_1)) \\
&\ge \psi(f(S_2), f(S_1 \cup \{e\}) - f(S_1)) \qquad (\because f(S_1) \le f(S_2)) \\
&\ge \psi(f(S_2), f(S_2 \cup \{e\}) - f(S_2)) \\
&\qquad (\because f(S_1 \cup \{e\}) - f(S_1) \ge f(S_2 \cup \{e\}) - f(S_2)) \\
&= \phi(f(S_2) + f(S_2 \cup \{e\}) - f(S_2)) - \phi(f(S_2)) \\
&= \phi(f(S_2 \cup \{e\})) - \phi(f(S_2)).
\end{aligned}
$$

This inequality is exactly

$$
h(S_1 \cup \{e\}) - h(S_1) \ge h(S_2 \cup \{e\}) - h(S_2),
$$

which shows that $h$ is also submodular. $\qquad\qquad\square$

**Lemma 7** (Lemma 1 in Lee & Oh (2024)). *Suppose that Assumption 1 hold. For any $\delta \in (0, 1]$, if we set $\eta = \frac{1}{2}\log(K+1) + 2, \Lambda = 84\sqrt{2}(d+1)\eta$, and $\alpha_t = \mathcal{O}(\sqrt{d+1}\log t \log K)$, then we have*

$$
\mathbb{P}\left(\forall t \ge 1, \|\mathbf{w}_t - \mathbf{w}^*\|_{\mathbf{H}_t} \le \alpha_t\right),
$$

*where the estimated parameter updated by the rule in Eq.(3).*

**Lemma 8** (Theorem 3.1 in Tropp (2011)). *Let $\mathcal{H}_1 \subset \mathcal{H}_2 \cdots$ be a filtration and consider a finite sequence $\{X_k\}$ of positive semi-definite matrices with dimension $d$ adapted to this filtration. Suppose that $\lambda_{\max}(X_k) \le R$ almost surely. Define the series $Y \equiv \sum_k X_k$ and $W \equiv \sum_k \mathbb{E}[X_k | \mathcal{H}_{k-1}]$. Then for all $\mu \ge 0, \gamma \in [0, 1)$ we have*

$$
\mathbb{P}[\lambda_{\min}(Y) \le (1-\gamma)\mu \text{ and } \lambda_{\min}(W) \ge \mu] \le d\left(\frac{e^{-\gamma}}{(1-\gamma)^{1-\gamma}}\right)^{\mu/R}.
$$

