# OpenReview forum: "Diversified Multinomial Logit Contextual Bandits"
_ICLR.cc/2026/Conference — ICLR 2026 Poster_

### Official Review · Reviewer_erMD · 2025-10-25

**Soundness:** 3
**Presentation:** 2
**Contribution:** 2
**Rating:** 4
**Confidence:** 4

**Summary:**

The paper proposes Diversified Multinomial Logit (DMNL) Contextual Bandits, extending the Multinomial Logit (MNL) choice model to explicitly promote diversity in the chosen assortments. The authors incorporate a submodular diversity function directly into the choice probability formulation. Finding optimal assortment in DMNL is intractable and requires exhaustive search. To tackle with this difficulty, the authors developed a UCB-based method that achieves a (1-1/(e+1)) approximate regret guarantee. The claimed theoretical result parallels known regret bounds for standard MNL bandits under uniform revenue assumptions.

**Strengths:**

The DMNL is a novel model that directly incorporates diversity as a submodular function added to the choice probability. This integration is conceptually appealing, as it allows diversity to influence user choice in a principled way rather than as a post-hoc balancing approach, seen in existing work.

To approximately solve the assortment selection problem, the authors constructed a UCB-based item-wise greedy algorithm that estimates the unknown relevance utility and diversity parameters jointly, avoiding exhaustive enumeration. The joint estimation of both parameter sets is a notable contribution. The authors further prove a regret bound of the algorithm that achieves comparable performance to existing MNL bandits, despite the added complexity of estimating diversity parameters.

**Weaknesses:**

The main weakness lies in the technical depth of the paper. While the formulation of the DMNL model is conceptually novel, the theoretical contributions are somewhat incremental. The improved approximation rate is a relatively straightforward derivation from the structure of the submodular function, which should be stated more of a proposition rather than theorem.
The item-size optimistic construction largely follows from UCB based algorithms that are commonly used in semi-bandit combinatorial multi-armed bandit literature. Although the adaptation to the DMNL context is sound, it seems not introducing new insights beyond existing frameworks. The joint estimation of $\theta$ and $\lambda$ comes from a relatively simple reparameterization of the choice probability that effectively treating diversity as an additional feature dimension. In addition, the presentation could be improved. The novelty is somewhat obscured by lengthy discussions that follow definitions and proofs, which destroys the flow of the paper.

**Questions:**

1. For the regret upper bound, is it matching in all parameters? It would be helpful to discuss how the additional diversity parameter introduces looser constraints.
2. A discussion on lower bound is also desirable.
3. What is the exact setup of the numerical experiments shown in Figure 2? A brief introduction would be desirable. Providing this context would make the figure much easier to understand without referring back to the appendix.

---

> ### Author Response · Authors · 2025-11-19
> **Rebuttal by Authors [1/4]**
>
> Thank you very much for taking the time to review our paper and for providing thoughtful feedback. We especially appreciate your positive comments regarding the novelty of the DMNL model, the joint estimation approach, and the strength of our theoretical results.
>
> At the same time, with all due respect, we feel that the current review may not fully capture the main contributions of our work, and we are grateful for the opportunity to clarify and further highlight the significance of these contributions, as well as for the valuable opportunity to communicate with the reviewer. We hope that, in light of the clarifications below, the contributions of our work can be considered with an open mind and evaluated on their own merits. We would be happy to address any further questions.
>
> Below, we address each of the reviewer's comments and questions in detail:
>
> ---
>
> ## Approximation rate
>
> The reviewer previously had wrtten:
> > **Reviewer erMD:** “The improved approximation rate is a relatively straightforward derivation from the structure of the submodular function, which should be stated more of a proposition rather than theorem.”
>
> With all due respect, the improved approximation rate **does not** follow straightforwardly from the structure of the submodular objective. Since the expected DMNL reward is submodular, one would naturally expect the existing $(1-1/e)$-approximation rate to hold (Nemhauser et al., 1978; Feige, 1998). However, Theorem 1 shows that, by exploiting structural properties unique to the DMNL model—specifically, the LogSumExp representation together with submodularity—we can surpass this classical limit and obtain a strictly tighter approximation rate. We believe that this is far from a straightforward derivation, and we regard this step as a necessary foundation for the subsequent item-wise optimistic construction analysis. Nevertheless, if the reviewer considers that framing it as a proposition--rather than a theorem--would provide a clearer structure, we are entirely willing to adopt that suggestion. That said, we strongly believe that the choice between labeling this important result as a theorem or as a proposition does not diminish the significance of our contributions.
>
> ---

---

> ### Author Response · Authors · 2025-11-19
> **Rubuttal by Authors [2/4]**
>
> ---
>
> ## Item-wise optimistic construction
>
> > **Reviewer erMD:** “The item-size optimistic construction largely follows from UCB based algorithms that are commonly used in semi-bandit combinatorial multi-armed bandit literature.”
>
> As we present our responses to this review comment, we would like to sincerely and respectfully emphasize two important points.
>
> (i) First, we note that being based on a UCB-style optimistic principle does not, by itself, render a contribution less significant. A vast body of work in the bandit literature builds on the UCB (or OFU) principle while introducing new modeling aspects, structural insights, or delicate analyses, and many of these works are widely regarded as substantial advances within their respective problem settings. Likewise (and arguably even more so in our setting, as we elaborate below), we adopt a UCB-style approach that is specifically tailored to our newly proposed DMNL framework and requires overcoming several novel technical challenges.
>
> (ii) More importantly, while our method follows a UCB-style optimistic principle, its construction differs substantially from standard semi-bandit UCB algorithms. And, we are happy to elaborate more below.
>
> In our problem setting, the feedback structure is entirely different from semi-bandits. In semi-bandit settings (e.g., Chen et al., 2013; Qin et al., 2014; Chen et al., 2016; Hwang et al., 2023), the agent receives individual and *independent* feedback for each item included in the super action (a subset of items). In contrast, the DMNL setting provides only a single choice feedback for the entire assortment which are dependent on each other, and *no direct feedback on item-level relevance or diversity is observed*. This makes the estimation of the relevance and diversity parameters more challenging.
>
> Furthermore, the mechanism through which optimism of the item set is achieved is also fundamentally different. In semi-bandit algorithms, once item-wise UCBs are *independently estimated*, optimism for the item set follows from *reward monotonicity together with access to a combinatorial optimization oracle*. In the DMNL setting, however, the optimistic value of an item depends on the current partial assortment, since the UCB bonus involves the set-dependent diversity feature. To be specific, the item-wise optimistic construction in Eq.(6) is equivalent to:
>
> $a_k=\arg\max_{a \in S^{(k-1)}} \sum_{i \in S^{(k-1)} \cup \\{a\\}} \exp (\[\mathbf{x_i}, g(S^{(k-1)} \cup \\{a\\})\]^\top \hat{\mathbf{w}} + \alpha \|\| \[\mathbf{x_i}, g(S^{(k-1)} \cup \\{a\\})\] \|\|_{\mathbf{H}^{-1}} )  $
>
> where $S^{(k-1)} := \\{a_1, \ldots, a_{k-1} \\}$.
>
> This shows that *the UCB for each candidate item must be evaluated dynamically, depending on how the current assortment is constructed*. This behavior does not arise in standard semi-bandit UCB algorithms, where item-wise UCBs are fixed once computed. Finally, our method does not require a combinatorial optimization oracle, which semi-bandit algorithms routinely rely on to ensure optimism of the selected set.
>
> Consequently, the proposed item-wise optimistic construction is not a direct adaptation of semi-bandit UCB methods, but a mechanism tailored for a much more restrictive feedback regime, where neither item-level feedback nor optimization oracles are available.
>
> ---

---

> ### Author Response · Authors · 2025-11-19
> **Rubuttal by Authors [3/4]**
>
> ---
>
> ## Joint estimation
>
> > **Reviewer erMD:** “The joint estimation of $\theta$ and $\lambda$ comes from a relatively simple reparameterization of the choice probability that effectively treating diversity as an additional feature dimension.”
>
> While the reparametrization allows the joint estimation of $\theta$ and $\lambda$ efficiently, we strongly believe that this step is by no means trivial, and the subsequent analysis requires more involved work. The joint confidence radius for $(\theta^\*, \lambda^\*)$ is often not tight enough for $\lambda^\*$; a loose confidence bound for $\lambda^\*$ fails to guarantee the submodularity of $\tilde{R}_t$, thereby preventing the item-wise optimistic construction from achieving its approximation guarantee.
>
> To overcome this issue, we introduce an adaptive uniform exploration phase. This phase ensures that the joint confidence width shrinks below the threshold at which the LogSumExp-UCB $\tilde{f}$ defined in Eq.(8) becomes monotone and submodular. Only after this point does the reparameterization correctly support the item-wise optimistic construction. Importantly, as established in Lemma 4, the length of this exploration phase is only $\mathcal{O}(\sqrt{d}\log T)$, so it contributes only a lower-order term to the overall regret.
>
> Thus, the key technical contribution is not the reparameterization itself, but *designing an exploration mechanism that preserves the improved approximation rate of the item-wise optimistic construction and ensures that the jointly estimated $(\hat{\theta}, \hat{\lambda})$ does not degrade regret*. This property is inherent to the DMNL framework and is fundamentally different from what arises in existing MNL bandits, requiring additional algorithmic design beyond simple reparameterization.
>
>
> ----
>
>
> ## Presentation
>
> > **Reviewer erMD:** “The presentation could be improved. The novelty is somewhat obscured by lengthy discussions that follow definitions and proofs, which destroys the flow of the paper.”
>
> Thank you for the comment. Because our analysis operates in a new framework that differs substantially from standard MNL bandits, we aimed to present the key ideas and technical components as clearly as possible, even if this required additional explanations. If there are specific parts where the presentation felt obscure or overly long, we would greatly appreciate the pointers. We are happy to clarify them in the discussion and reflect the improvements in the revision.
>
> ---

---

> > ### Author Response · Authors · 2025-11-19
> > **Rebuttal by Authors [4/4]**
> >
> > ---
> >
> > ## [Q1] Upper bound
> >
> > Thank you for the question. Since it is not specified what benchmark our regret bound is to be matched against, we believe that the reviewer is asking how our bound compares to the regret rates of existing MNL bandits and how the additional diversity parameter affects the regret. We provide our response under this interpretation.
> >
> > The leading term in our regret bound essentially matches the nearly minimax-optimal rate for MNL bandits with uniform revenues (Lee \& Oh, 2024), with the only difference being the natural dimensionality increase arising from estimating the additional diversity parameter $\lambda^\*$. Since the augmented parameter vector has dimension $d+1$, this shift is unavoidable rather than a loosened constraint.
> >
> > Regarding the effect of the diversity parameter on regret, as explained in our comment on [Joint estimation], $\theta^\*$ and $\lambda^\*$ are estimated jointly through the diversity-augmented features, and their uncertainties cannot be controlled independently. Ensuring the optimism required for the item-wise optimistic construction therefore necessitates an adaptive uniform exploration phase. As shown in Lemma 4, this additional exploration incurs only $\mathcal{O}(\sqrt{d}\log T)$ regret, contributing merely an additive logarithmic-order term to the overall regret.
> >
> >
> > ---
> > ## [Q2] Lower bound
> >
> > Thank you for giving us the opportunity to further strengthen our work. **In response to the reviewer’s suggestion, we have added the corresponding lower-bound result and its proof in Appendix F**. The lower bound we obtain is $\Omega({d\sqrt{T/K} })$, which matches the known lower bound for MNL bandits (Lee and Oh, 2024). Moreover, the leading term of the regret upper bound of our algorithm matches this lower bound in its dependence on the key parameters $d$, $K$, and $T$, up to logarithmic factors. Consequently, our algorithm is nearly minimax optimal for the DMNL bandit setting.
> >
> > Technically, we construct a discrete set of problem instances and show that, for any given policy, the expected worst-case regret over this set is bounded below. This implies that there must exist at least one instance whose regret is $\Omega({d\sqrt{T/K}})$. In our setting, the choice probabilities depend on the value of the assortment’s diversity function, and therefore the existing lower-bound arguments for MNL bandits cannot be applied directly. Specifically, we consider a non-constant, strict submodular $g_t$ and derive an inequality for the instantaneous regret lower bound, even though the optimal assortment includes an item that is not individually optimal in terms of their relevance scores. For this reason, the lower-bound analysis we develop for the DMNL bandit constitutes an independent and nontrivial contribution.
> >
> >
> > ----
> >
> >
> > ## [Q3] Experimental setup
> >
> > For each instance and round, the context features are independently drawn from a Gaussian distribution $\mathcal{N}(\mathbf{0}_d, \mathbf{I}_d)$ and clipped to the range $[-1/\sqrt{d},, 1/\sqrt{d}]^d$. Each item is also assigned a category, with the total number of distinct categories set equal to $K$, the size of the maximum assortment. The diversity function on an assortment $S$ is then defined as the exponential decaying categorical function (Example 1). To assess how effectively the proposed algorithm adapts to different relevance–diversity trade-offs, we fix the diversity parameter $\lambda^\*$ at several values. We then sample the relevance parameter $\theta^\*$ from a uniform distribution over $[-1/\sqrt{d}, 1/\sqrt{d}]^d$ and scale it such that $\|\| \theta^\* \|\|_2 + \lambda^\* = 1$. For each configuration, we conducted 10 independent runs, and all reported results are averaged over these runs.
> >
> > Due to page-limit constraints, we initially placed the experimental setup in Appendix G.1, along with detailed descriptions of $\texttt{OFU-MNL-DR}$ and $\texttt{OFU-DMNL-FULL}$, additional baselines we proposed. In light of the reviewer’s suggestion, we plan to add a brief description of the setup to the main paper where space permits after the rebuttal process.
> > Thank you for the suggestion.
> >
> > ---

---

### Official Review · Reviewer_wr1X · 2025-11-01

**Soundness:** 2
**Presentation:** 3
**Contribution:** 2
**Rating:** 6
**Confidence:** 4

**Summary:**

The paper proposes DMNL, an extension of the MNL choice model with a diversity parameter $g_t(S)$, where the diversity of the assortment in some sense modifies the probability of the outside option. A major challenge of this new model is that it is not possible to "greedily" solve the offline best assortment problem. As a result, the authors present $\gamma-$regret (approximate regret bound)

**Strengths:**

The paper is overall well-written and easy to understand. The model of DMNL is novel and bridges the gap between MNL choice model and diversity-focused assortment

**Weaknesses:**

1. I hope the authors can motivate the model better. Modeling the diversity via the submodular function $g_t$ seems a bit contrived. Is there a good understanding why should be the diversity of assortment be modelled in this way? e.g. Appendix C.2 gives some mathematical examples, can authors suggest if there are any examples where those functions may be appropriate?

2. The strict submodularity assumption is quite strong

3. Why is uniform exploration of lines 4-5 (Algorithm 1) required? What step of the proof fails without it?

4. Line 240 (Claim of "composition of a submodular function with a non-decreasing .....preserve submodularity" requires a proof or reference . it is not obvious.

5. What is utility of Theorem 1? In what scenarios/instances is it helpful to give a stronger constant in regret?

**Questions:**

1)$g_t$ needs to be sub-modular? It is not specified explicitly

also see the above weakness

---

> ### Author Response · Authors · 2025-11-19
> **Rebuttal by Authors [1/3]**
>
> Thank you for taking the time to review our paper and for providing thoughtful and constructive feedback. We sincerely appreciate your recognition of our contributions and the valuable comments you have shared. Below, we respond to each of your points in detail.
>
> ---
>
> ## [W1] Motivation of modeling diversity
>
> Thank you for the opportunity to elaborate on this.Our rationale is that many practically used diversity measures in assortment and recommendation systems naturally exhibit diminishing returns, which is exactly what submodularity encodes. For example, functions that count the number of distinct categories or genres in an assortment, measure coverage of item attributes (e.g., brands or styles), or quantify dispersion in an embedding space (e.g., via pairwise distances or spectral quantities of a Gram matrix) all have the property that adding an item that is similar to those already present contributes less incremental "diversity value" than adding an item from a new category or a far region in feature space. Such measures are monotone and (strictly) submodular by construction, so submodular functions provide a unifying and behaviorally plausible abstraction for a broad class of real-world diversity notions.
>
> Within this framework, our contribution is to study how a given diversity function $g_t$ interacts with user choice behavior and to design learning algorithms that can efficiently estimate the strength of this diversity effect and construct near-optimal assortments with provable approximation and regret guarantees. We will revise the main text to make this modeling motivation more explicit and to better connect the examples in Appendix C.2 to concrete application scenarios.
>
>
> ---
>
> ## [W2] Strict submodularity
>
> We are glad to address this comment. Importantly, the strict submodularity assumption applies only to the diversity function $g_t$, which is fully specified by the system and computed directly from item context features. In our model, user choice probabilities depend on a known diversity score $g_t$ and an *unknown* parameter $\lambda^*$. As discussed in Appendix C.2, widely used diversity measures—such as categorical functions with exponentially decaying increments and categorical-level coverage functions—naturally satisfy strictly diminishing returns. Since $g_t$ is derived deterministically from item metadata, strict submodularity is an internal modeling choice fully under the agent’s control, and *does not restrict user behavior*.
>
> By contrast, prior work on combinatorial bandits or submodular bandits typically relies on assumptions involving *external information the agent cannot control*, such as (i) a black-box combinatorial optimization oracle, or (ii) intermediate marginal-gain feedback during set construction. Such assumptions require the environment to reveal information that is not available in the DMNL setting, where the agent observes only a single user choice and never receives separate direct feedback on $g_t(S)$ or on $\lambda^*$.
>
> Our contribution is to show that strict submodularity provides a structural condition that replaces these external requirements: it ensures that the optimistic reward $\tilde{R}_t$ remains submodular under bandit feedback, enabling an item-wise optimistic construction without any oracle access or intermediate feedback. From this perspective, strict submodularity is not a stronger assumption, but rather a practical and controllable alternative to the external feedback and oracle assumptions used in prior combinatorial and submodular bandit frameworks. It provides a principled mechanism to achieve tractable learning under extremely limited feedback, which is central to our theoretical guarantees.
>
> ---

---

> ### Author Response · Authors · 2025-11-19
> **Rebuttal by Authors [2/3]**
>
> ---
>
> ## [W3] Uniform exploration
>
> As explained in the main paper (lines 338–346), the uniform exploration is required to ensure that the confidence widths for $\lambda^\*$ become sufficiently small before the item-wise optimistic construction is applied. Because $\hat{\theta}$ and $\hat{\lambda}$ are estimated jointly through the diversity-augmented feature representation, their individual uncertainties cannot be estimated separately. As a result, without adaptive uniform exploration, the confidence bound for $\lambda^\*$ can remain loose.
>
> In more detail, the approximation guarantee of the item-wise optimistic construction relies on the monotonicity and submodularity of LogSumExp–UCB function $\tilde{f}_t$ defined in Eq. (8). However, when the confidence radius of the diversity parameter is large—equivalently, when the condition that triggers the adaptive exploration phase is not met—$\tilde{f}_t$ cannot be ensured to be submodular without any additional assumptions. In this case, the output of the item-wise optimistic construction $\tilde{R}_t(S_t)$ cannot be meaningfully compared to the exhaustive maximizer $R(S_t^*, \theta^\*, \lambda^\*)$, and its approximation guarantee breaks down.
>
> The uniform exploration phase resolves this issue. After this phase, Lemma 1 (Appendix D) shows that the confidence widths shrink sufficiently so that $\tilde{f}_t$ becomes both monotone and submodular. This structural property is essential for our analysis: it allows us to leverage the submodularity of $\tilde{f}_t$ to relate the item-wise optimistic construction to the optimal assortment. Moreover, as shown in Lemma 4, the number of uniform-exploration rounds is only $O(\sqrt{d}\log T)$, so this phase is short and does not materially affect the overall regret.
>
>
> ---
>
>
> ## [W4] Preservation of submodularity
>
> Thank you for the suggestion.
> The fact that the composition of a monotone submodular function with a non-decreasing concave function preserves submodularity is a known result in the submodular optimization literature (Proposition B.6 in F. Bach (2013)). We will add an explicit citation in the revised version.
>
> Here, we provide a brief proof.
> Let $f$ be a monotone submodular function and $\phi$ a non-decreasing concave function. Define $h(S) := \phi(f(S))$. By submodularity of $f$, for any $S_1 \subseteq S_2$ and $e \notin S_2$, we have:
>
> $
> f(S_1 \cup \{e\}) - f(S_1)
> \ge
> f(S_2 \cup \{e\}) - f(S_2).
> $
>
> Also, concavity and the non-decreasing property of $\phi$ imply that the difference $\psi(x, \delta) = \phi(x+\delta) - \phi(x)$ is non-increasing in $x$ and non-decreasing in $\delta$.
> That is, if $x_1 \le x_2$, then $\psi(x_1, \delta) \ge \psi(x_2, \delta)$, and if $\delta_1 \le \delta_2$, then $\psi(x, \delta_1) \le \psi(x, \delta_2)$.
>
> Using this, we obtain:
>
> $
> \phi(f(S_1 \cup \{e\})) - \phi(f(S_1)) \\
> $
>
> $
> = \psi(f(S_1), f(S_1 \cup \{e\}) - f(S_1)) \\
> $
>
> $\ge \psi(f(S_2), f(S_1 \cup \{e\}) - f(S_1)) \\
> \quad (\because f(S_1) \le f(S_2)) \\
> $
>
> $
> \ge \psi(f(S_2), f(S_2 \cup \{e\}) - f(S_2)) \\
> \quad (\because f(S_1 \cup \{e\}) - f(S_1) \ge f(S_2 \cup \{e\}) - f(S_2)) \\
> $
>
> $
> = \phi(f(S_2 \cup \{e\})) - \phi(f(S_2))
> $
>
>
> This inequality is exactly
>
> $
> h(S_1 \cup \{e\}) - h(S_1)
> \ge
> h(S_2 \cup \{e\}) - h(S_2),
> $
>
> which shows that $h$ is also submodular.
>
> **References**
>
> F.Bach. (2013), "Learning with submodular functions: A convex optimization perspective."
>
>
> ---
>
> ## [W5] Utility of Theorem 1
>
> For general monotone submodular maximization, the best-known approximation rate guarantee for item-wise greedy construction is $1 - 1/e$ (Nemhause et al., 1978), and improving this bound is known to be \textit{intractable} (Feige, 1998). Since the expected reward of a DMNL assortment is also submodular, one would naturally expect this $(1 - 1/e)$-approximation rate to apply the item-wise greedy construction as well. However, Theorem 1 shows that, due to structural properties specific to the DMNL model, the item-wise greedy construction in fact enjoys a *strictly tighter approximation rate*. Thus, Theorem 1 goes beyond what is achievable for arbitrary submodular functions and demonstrates that the *DMNL structure enables a strictly better guarantee*.
>
> This improved approximation factor is particularly valuable because our algorithm operates without a *combinatorial optimization oracle* or *intermediate marginal-gain feedback*—both of which are commonly assumed in combinatorial or submodular bandits but are unavailable in the DMNL setting. Since the regret bound scales directly with the approximation factor, achieving a stronger approximation rate under such limited feedback is crucial for both theoretical and practical performance. **Theorem 1 therefore provides a more effective and theoretically grounded assortment construction method precisely in a setting where traditional submodular bandit tools cannot be applied.**
>
> ---

---

> > ### Author Response · Authors · 2025-11-19
> > **Rebuttal by Authors [3/3]**
> >
> > ---
> > ## [Q1] submodular diversity
> >
> > In our analysis, we assume $g_t$ to be monotone and $w$-strict submodular, which is stated explicitly in Assumption 3 and Theorem 2.
> >
> > ---

---

> > > ### Comment · Reviewer_wr1X · 2025-11-25
> > > **Thanks for the rebutal**
> > >
> > > Thanks for carefully answering my question. Many of my queries have been resolved. Please revise the main text to incorporate the discussions and explanations outlined above, as needed.

---

### Official Review · Reviewer_KWsv · 2025-11-01

**Soundness:** 3
**Presentation:** 3
**Contribution:** 3
**Rating:** 6
**Confidence:** 3

**Summary:**

Proposes a diversified MNL (DMNL) contextual bandit: standard MNL utilities plus a submodular diversity term; gives a greedy OFU algorithm with an improved approximation constant over 1−1/e and √T-type regret; synthetic results show better relevance–diversity trade-offs than MNL baselines.

**Strengths:**

- Bridges choice modeling and diversity via DMNL; captures the relevance–diversity tension within the click probabilities rather than via ad-hoc reward shaping.
- Item-wise optimistic greedy avoids black-box combinatorial oracles yet has provable guarantees

**Weaknesses:**

- The algorithm takes \(g_t(S)\) as given. Many applications may only have noisy diversity signals; robustness to misspecification or learned $g_t$ is not explored.
- Experiments appear synthetic; adding a real recommendation/assortment dataset would strengthen the contribution

**Questions:**

- Is \(g_t(S)\) assumed exactly known at decision time? How would OFU-DMNL adapt if \(g_t\) is observed with noise or must be learned from implicit feedback?
- Can guarantees be given under submodularity (ω=0) ?

---

> ### Author Response · Authors · 2025-11-19
> **Rebuttal by Authors [1/2]**
>
> Thank you for taking the time to review our paper and for providing thoughtful and constructive feedback. We sincerely appreciate your recognition of our contributions and the valuable comments you have shared. Below, we respond to each of your points in detail.
>
> ---
>
> ## [W1 \& Q1] Given $g_t$ does not mean knowing how diversity affects choices
>
> We are happy to address this question. In our setting, we work with a given diversity function $g_t$, which is often readily available in applications. Typical examples include the number (or coverage) of *distinct categories* of items in an assortment, the coverage of particular features such as *genres* or *brands*, or, more algebraically, the dispersion of item embeddings (e.g., the minimum eigenvalue of the Gram matrix formed by the item features in an assortment $S$). In all of these examples, $g_t(S)$ is a deterministic function of the items included in the assortment and can be evaluated once a candidate set $S$ is specified. (Of course, we do not know which assortment is optimal a priori.)
>
> Crucially, however, knowing $g_t$ is very different from knowing how diversity affects user choice behavior. The algorithm does not know in advance which items (or assortments) are optimal in terms of diversity and/or relevance, and, more importantly, it does **not** know a priori how this diversity information should be traded off against relevance in the choice probabilities. This trade-off is governed jointly by the **unknown** parameters $\lambda^\*$ and **$\theta^\*$**, and must be inferred from noisy bandit feedback: in each round, the agent observes only a single user choice from the offered assortment, and never receives separate direct feedback on $g_t(S)$ or on $\lambda^\*$.
>
> Thus, even when $g_t$ itself is given, its **effect** on user choice probabilities—and the joint optimization of diversity and relevance to maximize reward—must still be learned online. This is the central learning challenge in the DMNL bandit setting and makes the problem substantially more difficult than standard MNL bandits or even the problem of optimizing diversity only. That said, extending our framework to possibly noisy $g_t$ (if carefully defined) settings would be interesting direction for future work, but does not diminish the core contributions established under the current assumption of given $g_t$ that is well motivated by application examples.
>
> Our contribution is to provide a principled solution to precisely this challenge: we show that, even under such indirect feedback, one can still jointly estimate $(\theta^\*, \lambda^\*)$ and construct near-optimal assortments with provable approximation and regret guarantees without relying on computation oracle. We believe this addresses a fundamentally difficult aspect of diversity-aware choice modeling that has not been handled in prior work.
>
> ---

---

> ### Author Response · Authors · 2025-11-19
> **Rebuttal by Authors [2/2]**
>
> ---
>
> ## [W2] Experiments
>
> First, there is no publicly available dataset suitable for evaluating assortment bandit algorithms—even for the single-item selection case—without conducting a live online field experiment. Consequently, most prior work (Ou et al., 2018;  Oh \& Iyengar, 2021; Perivier \& Goyal, 2022; Zhang \&Sugiyama, 2024; Lee \& Oh, 2024; 2025) has relied on synthetic data.
> To partially bridge this gap, we additionally designed a semi-synthetic experiment based on a real-world dataset, which provides the most practical and feasible alternative in the absence of access to online field deployment.
>
> We used the Massive Rotten Tomatoes Movie \& Review dataset ([link](https://www.kaggle.com/datasets/andrezaza/clapper-massive-rotten-tomatoes-movies-and-reviews)), which contains over $1.4$M+ reviews on $140$K+ unique movies, each labeled as positive or negative, along with rich movie-level metadata. We first converted each review text into a vector representation using TF-IDF (via TfidfVectorizer from scikit-learn), followed by dimensionality reduction using truncated SVD to obtain $d$-dimensional context vectors. This process resulted in a context–label dataset suitable for downstream modeling.
>
> From this dataset, we trained a linear model to classify the binary labels, which was then used to approximate the true relevance utility of each movie from its context features.
> This constructed an online assortment selection environment in which we evaluated the performance of standard MNL bandit baselines, their variants, and our proposed method.
>
> In each round of the online experiment, we randomly sampled $N$ movies, and asked the algorithm to choose an assortment of size $K$. We use exponential decaying categorical function defined as $g(S):=1+\rho+\ldots + \rho^{n_S-1}$, where $n_S$ is the number of categories covered by the assortment $S$. The experimental results (cumulative regret at time step $T=10,000$) are summarized in the table below and detailed in Appendix G.3.
>
> |$N$|$K$|$d$|$\texttt{OFU-MNL+}$|$\texttt{OFU-MNL-DR}$|$\texttt{OFU-DMNL}$ (ours)|
> |---|---|---|---|---|---|
> |$30$|$5$|$5$|$73.16 \pm 30.93$ | $43.18 \pm 13.54$ | $\textbf{28.67}$ $\pm$ $\textbf{11.99}$|
> |$30$|$5$|$10$|$50.97 \pm 14.55$ | $39.25 \pm 17.65$ | $\textbf{30.07}$ $\pm$ $\textbf{12.99}$|
> |$50$|$10$|$5$|$25.60 \pm 17.92$ | $22.03 \pm 13.13$ | $\textbf{16.52}$ $\pm$ $\textbf{7.33}$|
> |$50$|$10$|$10$| $12.16 \pm 5.12$ |  $10.33 \pm 5.34$ | $\textbf{8.34}$ $\pm$ $\textbf{4.49}$|
>
>
>
> ---
>
> ## [Q2] Strict submodularity
>
> We are happy to address this question. The key analytical difficulty in our setting is ensuring the approximate optimism of the assortment constructed by the item-wise optimistic construction, i.e., $R(S_t^*, \theta^\*, \lambda^\*) \le \gamma \tilde{R}_t(S_t)$. To guarantee this, it suffices to show that the LogSumExp–UCB function $\tilde{f}_t$ defined in Eq. (8) is both monotone and submodular.
>
> However, as explained in Appendix C.1, $\tilde{f}\_t$ is not submodular in general, even when $g_t$ itself is submodular. The difficulty arises because the UCB bonus term contains the convex norm $\\|z_{ti}(S)\\|_{H^{-1}}$, whose marginal increase does not necessarily diminish as the set expands. This breaks the diminishing-returns structure and hence the submodularity of $\tilde{f}_t$.
>
> The strict submodularity of the diversity function $g_t$ provides a quantitative diminishing-returns margin that we leverage to control the UCB bonus term once the diversity parameter is estimated with a sufficiently small confidence width (Lemma 1). Establishing the resulting submodularity of $\tilde{f}_t$ under joint parameter uncertainty is technically delicate and constitutes one of the central challenges of our analysis.
>
> We believe that weakening the strict submodularity would require substantially more refined analytical techniques, which lie beyond the scope of this paper. Importantly, strict submodularity is imposed only on the system-defined diversity function $g_t$, which is entirely under the agent’s control and naturally satisfied by common diversity measures (Appendix C.2). Even under this controllable structural assumption—and without access to a black-box optimization oracle or intermediate marginal-gain feedback—we obtain strong approximation and regret guarantees.
>
> ---

---

### Meta-Review · Area_Chair_JGdx · 2026-01-05

**Summary:**

The reviewers agree this paper makes a solid and well-motivated contribution by introducing the diversified multinomial logit contextual bandit model, which combines relevance and diversity in one framework. The modeling choice is seen as new and appealing, and the proposed UCB-style algorithm looks technically sound, efficient, and well analyzed. Some reviewers initially had doubts about the strength of the assumptions and whether the theoretical novelty was deep enough, but the rebuttal helped a lot by clarifying the modeling motivation, adding a matching lower bound, and strengthening both theory and experiments. Overall, the work is considered a meaningful improvement over standard MNL bandits, with clear guarantees and convincing experiments, and seems like a good fit for ICLR.

**Reviewer Concerns:**

Most of the reviewers’ concerns were addressed in the rebuttal. The authors clarified the motivation for submodular diversity, explained the role of strict submodularity and uniform exploration, added missing references, and provided additional theoretical results including a lower bound. The lack of real-world experiments was partially addressed with a semi-synthetic dataset. Some minor concerns about presentation and strength of assumptions remain, but they are no longer blocking.

**Reviewer Scores:**

Reviewers who were initially above the threshold would likely keep their scores unchanged. Reviewers who were borderline or slightly negative mostly increased their scores after the rebuttal and clarifications. Overall, the discussion would lead to a clearer consensus in favor of acceptance.

---

### Decision · Program_Chairs · 2026-01-26

Accept (Poster)